# Axon-specific microtubule regulation drives asymmetric regeneration of sensory neuron axons

Ana Catarina Costa[1,2], Blanca R Murillo[1], Rita Bessa[1], Ricardo Ribeiro[1],
Tiago Ferreira da Silva[3], Patrícia Porfírio-Rodrigues[4], Gabriel G Martins[4],
Pedro Brites[3], Matthias Kneussel[5], Thomas Misgeld[6], Monika S Brill[6],
Monica M Sousa[1]*

[1]Nerve Regeneration Group, Instituto de Biologia Molecular e Celular (IBMC),
Instituto de Investigação e Inovação em Saúde (i3S), University of Porto, Porto,
Portugal; [2]Graduate Program in Molecular and Cell Biology, Instituto de Ciências
Biomédicas Abel Salazar, University of Porto, Porto, Portugal; [3]Neurolipid Biology
Group, Instituto de Biologia Molecular e Celular (IBMC), Instituto de Investigação e
Inovação em Saúde (i3S), University of Porto, Porto, Portugal; [4]Advanced Imaging
Unit, Instituto Gulbenkian de Ciência, Lisboa, Portugal; [5]Institute of Molecular
Neurogenetics, Center for Molecular Neurobiology Hamburg, ZMNH, University
Medical Centre Hamburg-Eppendorf, Hamburg, Germany; [6]Institute of Neuronal
Cell Biology, Technical University of Munich, Munich, Germany; Munich Cluster for
Systems Neurology (SyNergy), Munich, Germany

*For correspondence:
msousa@i3s.up.pt

Reviewing Editor: Kassandra
M Ori-McKenney, University of
California, Davis, United States

## eLife Assessment

In their **important** article, Costa et al. establish an in vitro model for dorsal root ganglion (DRG) axonal asymmetry, revealing that central and peripheral axon branches have distinct patterns of microtubule populations that are linked to their differential regenerative capacities. The authors employ creative tissue culture methods to demonstrate how these branches develop uniquely in vitro, offering a potential explanation for long-observed regeneration disparities. The **convincing** evidence provides a contribution to our understanding of the neuronal cytoskeleton and axonal regeneration.

**Abstract** Sensory dorsal root ganglion (DRG) neurons have a unique pseudo-unipolar morphology in which a stem axon bifurcates into a peripheral and a central axon, with different regenerative abilities. Whereas peripheral DRG axons regenerate, central axons are unable to regrow. Central axon regeneration can however be elicited by a prior conditioning lesion to the peripheral axon. How DRG axon asymmetry is established remains unknown. Here we developed a rodent in vitro system replicating DRG pseudo-unipolarization and asymmetric axon regeneration. Using this model, we observed that from early development, central DRG axons have a higher density of growing microtubules. This asymmetry was also present in vivo and was abolished by a conditioning lesion that decreased microtubule polymerization of central DRG axons. An axon-specific microtubule-associated protein (MAP) signature, including the severases spastin and katanin and the microtubule regulators CRMP5 and tau, was found and shown to adapt upon conditioning lesion. Supporting its significance, interfering with the DRG MAP signature either in vitro or in vivo readily abolished central-peripheral asymmetries in microtubule dynamics and regenerative ability. In summary, our data unveil that axon-specific microtubule regulation drives asymmetric regeneration of sensory neuron axons.

## Introduction

To execute their functions, neurons evolved a high degree of polarization (*Banker, 2018*). Sensory dorsal root ganglion (DRG) neurons show a unique pseudo-unipolar morphology in which a single stem axon bifurcates into a peripheral and a central axon, each with distinct functions and properties (*Nascimento et al., 2018*). The peripheral DRG axon is the site of action potential generation that is transmitted to the central axon, which synapses with second-order neurons. While the different environments in which the two DRG axons lie certainly play an important role in their distinctive features, axon-intrinsic properties have emerged. In this respect, peripheral DRG axons show larger diameters (*Suh et al., 1984*), higher axonal transport rates (*Komiya and Kurokawa, 1978*; *Wujek and Lasek, 1983*), and increased regenerative capacity (*Komiya, 1981*; *Wujek and Lasek, 1983*) when compared to central ones (reviewed in *Nascimento et al., 2018*).

Given the different competence of DRG axons to mount a regenerative response, DRG neurons have been widely used to study the regulation of axon regeneration. Interestingly, while DRG peripheral axons regrow following injury in vivo, the central axon only regenerates if a priming conditioning lesion to the peripheral axon is made (*Neumann and Woolf, 1999*; *Richardson and Verge, 1987*). This conditioning peripheral lesion induces molecular changes in DRG neurons enabling their central axons to acquire growth competence, even in an inhibitory environment (*Hoffman, 2010*; *Mar et al., 2014a*). We have previously shown that injury to the DRG peripheral axon induces a global increase in axonal transport that extends to the central axon, supporting axon regeneration (*Mar et al., 2014b*). Since axonal transport relies on microtubule-based motors, microtubule organization in the DRG axons may determine their biological asymmetry. However, in contrast to multipolar neurons where microtubule polarity governs asymmetric distribution of cargoes (*Tas et al., 2017*), the mechanisms underlying differential axonal transport rates in DRG neurons are not understood. An important setback in studying DRG neurons is that while different structural aspects of their two axons may have important implications in their function, most of the studies are restricted to the use of in vitro systems in which these neurons fail to recapitulate their pseudo-unipolar morphology (*Nascimento et al., 2018*).

Here we established an in vitro model replicating DRG pseudo-unipolarization and asymmetric axon development that is an important tool to understand the biology of sensory neurons and the mechanisms underlying optimal axon (re)growth. Combining this in vitro model with in vivo studies, we show that axon-specific regulation of microtubule polymerization underlies regeneration asymmetry of sensory neuron axons.

## Results

### DRG neurons in vitro undergo pseudo-unipolarization, replicating in vivo asymmetries and regenerative capacity

To investigate the cellular and molecular mechanisms underlying the physiological asymmetries and regenerative capacity of DRG neurons, it is crucial to develop a primary cell culture system that promotes physiological cellular behavior. DRG neurons from developing or adult rodents cultured in vitro often fail to fully mimic their in vivo cell biology, typically producing multiple, highly ramified neurites (*Nascimento et al., 2018*). Ideally, DRG neuron cultures should replicate the morphogenic process of pseudo-unipolarization while preserving the peripheral-central axonal asymmetries (*Nascimento et al., 2022*). In this respect, we developed an in vitro system where, in the presence of DRG glial cells, pseudo-unipolarization occurs. In this system, during the early phases of development, DRG neurons exhibit a spindle or eccentric-shaped bipolar morphology (*Figure 1A*, stage I), and as they mature the two axons move closer, adopting a bell-shaped bipolar morphology (*Figure 1A*, stage II), that later converts into a pseudo-unipolar morphology (*Figure 1A*, stage III). At DIV21, we observed 4 ± 1% multipolar (with more than two neurites), 35 ± 8% bipolar, 17 ± 5% bell-shaped, and 43 ± 3% pseudo-unipolar DRG neurons (*Figure 1B*). When we segmented pseudo-unipolar DRG neurons, the stem axon and the bifurcating axons were clearly observed (*Figure 1C*, *Figure 1—video 1*). While DRG neurons undergo pseudo-unipolarization (*Figure 1D*, *Figure 1—video 2*), a reduction in diameter (*Figure 1D–E*) coupled with an increased length of the stem axon (*Figure 1D and F*) occurs, mirroring in vivo development (*Matsuda et al., 2000*). To further understand stem axon formation, we tracked the displacement of the DRG T-junction (*Figure 1D*, black arrowhead) and of the top of the cell body (*Figure 1D*, green arrowhead) in relation to the initial

**eLife digest** When nerves in our body are damaged, their ability to repair themselves depends on where they are. Some nerves, like those in the arms and legs, can heal, while the ones in the spinal cord cannot. This difference is particularly striking in dorsal root ganglion (DRG) neurons, which have a unique structure. Unlike other neurons, which transmit signals along a single long projection called an axon, DRG neurons branch into two axons – one connecting to the body and the other to the spinal cord. While the branch leading to the body can heal, the one connecting to the spinal cord is unable to regenerate.

It is not clear how DRG neurons develop axons with these differing abilities. Researchers have found that an injury to the body side branch, known as the peripheral axon, can stimulate regrowth in the stretch leading to the spinal cord, known as the central axon. The damage increases the transport of molecules along both axons, boosting the repair of the whole neuron. This suggests that microtubules, the internal highways for transporting materials through cells, may contribute to the difference between the regenerative ability of the two axons of the DRG neuron.

To explore this, Costa et al. studied DRG neurons grown in the laboratory and rodents. Powerful microscopes revealed that the central axons contain more actively growing microtubules than the peripheral axons. However, when the peripheral axon was damaged, the central axon reduced microtubule growth, making it more capable of regeneration. Costa et al. also identified that injury caused changes in levels of microtubule-associated proteins (MAPs), which regulate microtubule behaviour. Reducing the amount of one of these MAPs prevented axon repair in both cell cultures and animal models.

These findings help explain why some nerve fibres regenerate while others do not, highlighting the role of microtubules in this process. Further research is needed to determine whether targeting MAPs may lead to new treatments for spinal cord injury or other nervous system damage in humans.

position of the T-junction (*Figure 1D*, red dot). After establishing the DRG T-junction, the cell body is displaced significantly to the opposite direction (*Figure 1D*, green arrowhead; and *Figure 1G*), while the position of the base of the stem axon remains relatively stable (*Figure 1D*, black arrowhead; and *Figure 1G*). This supports that cell body bulging participates in stem axon formation, as previously suggested (*Matsuda et al., 2000*). Interestingly, at early days in vitro (DIV 7–14), pseudo-unipolar DRG could be reverted to a bell-shaped bipolar morphology, while at later DIVs (DIV 14–21), this transition was never observed (*Figure 1—video 2*). In vivo studies have shown that peripheral and central DRG axons differ in diameter, with peripheral axons being larger (*Suh et al., 1984*). When measuring the two bifurcating axons 3–5 µm away from the DRG T-junction, we observed that pseudo-unipolar DRG neurons exhibit two axons with distinct diameters: a thin, central-like branch and a thicker, peripheral-like branch (*Figure 1H and I*). This axonal diameter asymmetry is established early in DRG development, as bipolar neurons already displayed axons of different diameters (*Figure 1H*).

To further support the relevance of our in vitro system, we explored whether it replicates the observed in vivo asymmetry in regenerative capacity. For that, either the large peripheral-like (*Figure 2A*, *Figure 2—video 1*) or the thin central-like axon (*Figure 2B*, *Figure 2—video 1*) were injured by laser axotomy and their regenerative response was followed. While the length of retraction following injury was the same for both DRG axons (*Figure 2C*), the duration of retraction was higher in central-like axons (*Figure 2D*), suggesting that these require more time to assemble a regenerative response (*Koseki et al., 2017*). Moreover, whereas over 50% of the injured peripheral-like axons were able to regenerate with an average length of 33.4 µm, only approximately 10% of injured central ones showed the same ability, with a 3.8-fold decreased length (8.7 µm) (*Figure 2E and F*).

In summary, we established an in vitro system that mirrors the physiological developmental stages of DRG neurons in vivo, recapitulating their regenerative asymmetries. This model is a valuable tool for studying the unique biology of DRG neurons as well as for investigating the mechanisms driving axon regeneration.

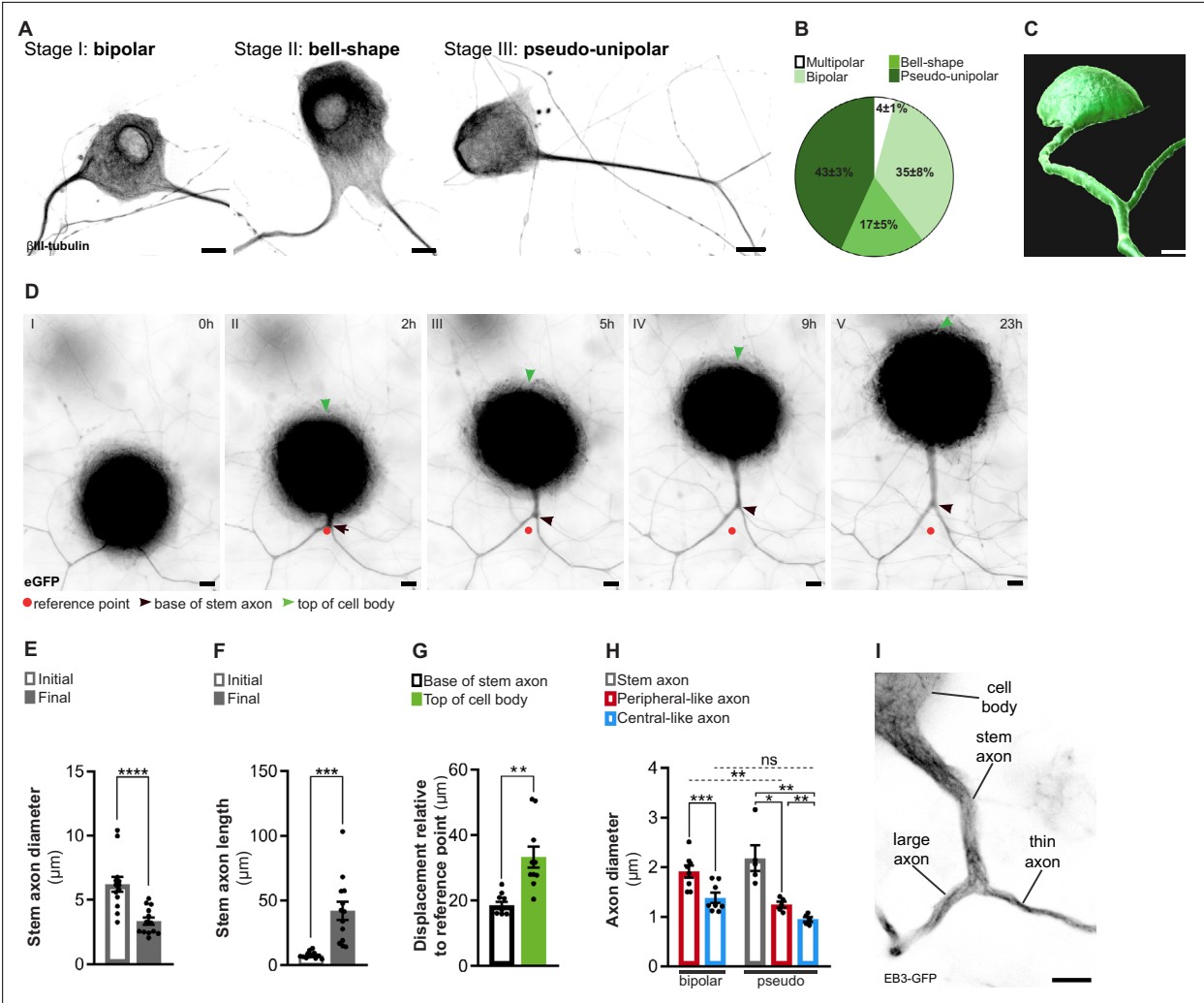

**Figure 1.** In vitro dorsal root ganglion (DRG) neurons recapitulate in vivo developmental stages. (**A**) In vitro DRG neurons labeled with βIII-tubulin depicting different development stages. Scale bar, 10 µm. (**B**) Percentage of different DRG neuron morphologies at DIV21 (n = 3 independent experiments, 100 cells per experiment). (**C**) Imaris segmentation of a pseudo-unipolar DRG neuron transduced with AAV-CMV-eGFP. Scale bar, 7 µm. (**D**) In vitro DRG neurons transduced with AAV-CMV-eGFP depicting stem axon formation. Scale bar, 10 µm. (**E, F**) Stem axon diameter (**E**) and length (**F**) of DRG neuron axons from the formation of the stem axon (Initial) to the final stage of pseudo-unipolarization (Final) (n = 13 neurons; paired t-test, diameter ****p<0.0001, length ***p=0.0004). (**G**) Stem axon and cell-body displacement during pseudo-unipolarization (n = 13 neurons; paired t-test, **p=0.0020). (**H**) In vitro diameter of DRG axons; n = 5–8 independent experiments, 5–10 neurons/experiment; paired t-test in bipolar neurons, ***p=0.0003; repeated measures (RM) one-way ANOVA in pseudo-unipolar neurons, stem-peripheral *p=0.0196, stem-central **p=0.0069, peripheral-central **p=0.0048; for comparisons amongst peripheral and central-like axons from bipolar and pseudo-unipolar neurons, a two-way ANOVA was used (peripheral: **p=0.0039; central: p=0.9829). (**I**) In vitro pseudo-unipolar DRG neuron transduced with the lentivirus CMV-EB3-GFP depicting different axon diameter. Scale bar, 5 µm. Data are represented as mean ± SEM.

The online version of this article includes the following video(s) for figure 1:

**Figure 1—video 1.** Segmentation of an in vitro pseudo-unipolar embryonic rat dorsal root ganglion (DRG) neuron expressing eGFP acquired using timelapse confocal microscopy.

https://elifesciences.org/articles/104069/figures#fig1video1

**Figure 1—video 2.** In vitro embryonic dorsal root ganglion (DRG) neurons assembling a stem axon.

https://elifesciences.org/articles/104069/figures#fig1video2

## DRG axons exhibit asymmetric microtubule polymerization, which is downregulated by a conditioning lesion

Early studies demonstrated that axonal transport is asymmetrically regulated in DRG axons, as peripheral axons exhibit higher transport rates (reviewed in *Nascimento et al., 2018*). Aligning with in vivo

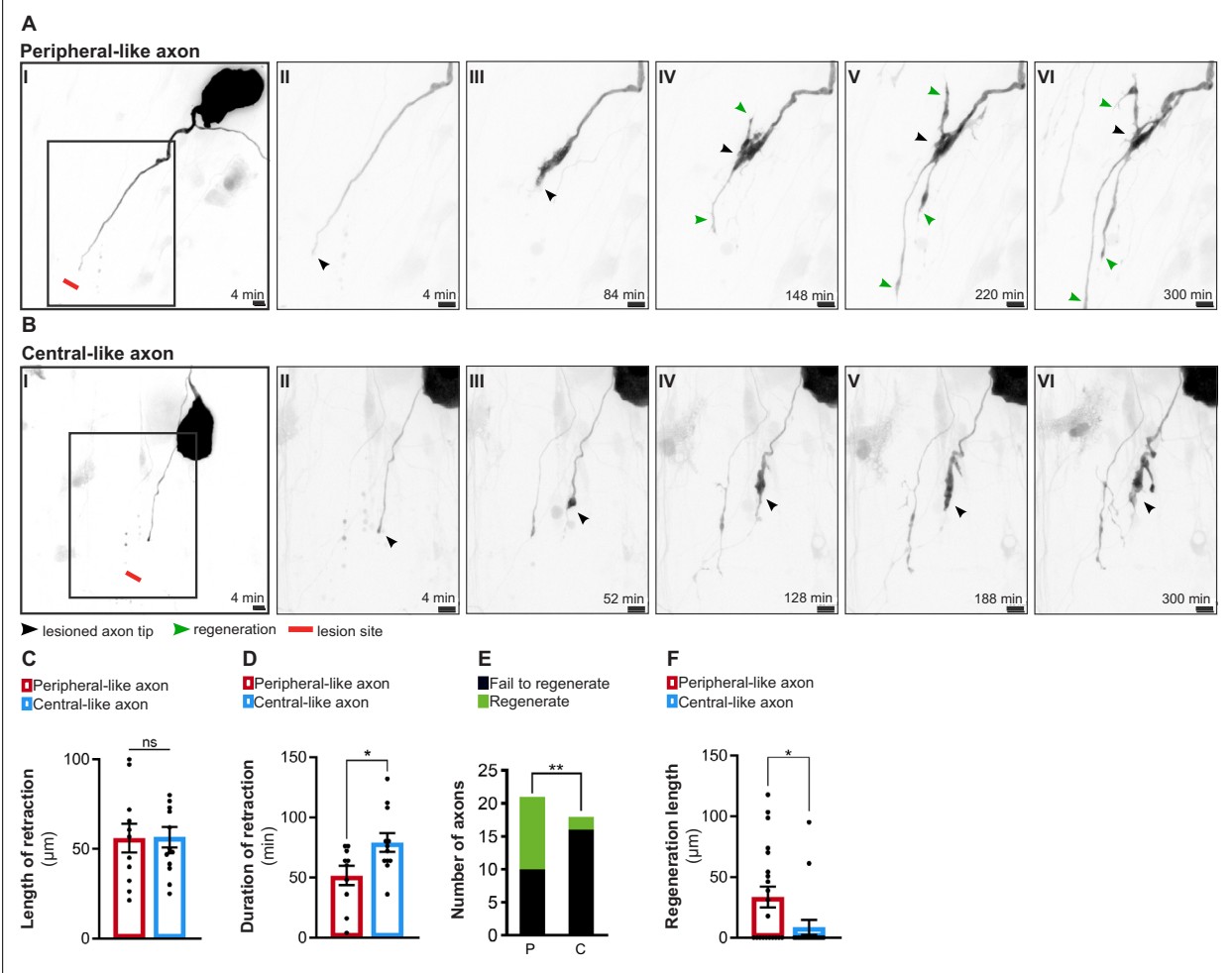

**Figure 2.** In vitro dorsal root ganglion (DRG) neurons recapitulate in vivo regenerative asymmetry. (**A, B**) Live imaging of in vitro regeneration of a peripheral-like (**A**) and a central-like axon (**B**). Scale bar, 10 μm. The injury sites are marked by red lines; lesioned axon tips by black arrowheads and regenerating axons by green arrowheads. (**C, D**) Length (**C**) and duration (**D**) of axon retraction after laser axotomy (n = 10–14 axons, three independent experiments; unpaired *t*-test, length p=0.9541, duration *p=0.0228). (**E**) Number of regenerating and non-regenerating peripheral-like (P) and central-like (C) DRG axons. The chi-square test was used (five independent experiments, **p=0.0082). (**F**) Regeneration length of peripheral and central-like axons following laser axotomy (n = 18–21 axons, five independent experiments; unpaired *t*-test, *p=0.0297). Data are represented as mean ± SEM.

The online version of this article includes the following video for figure 2:

**Figure 2—video 1.** Laser axotomy of peripheral and central-like rat dorsal root ganglion (DRG) axons.
https://elifesciences.org/articles/104069/figures#fig2video1

data, in our in vitro model, peripheral-like axons had a higher flux of anterogradely moving mitochondria than central-like axons (*Figure 3A and B*, *Figure 3—video 1*), further supporting the relevance of this system. Given the importance of microtubules in axonal transport (*Nirschl et al., 2017*), we next investigated how microtubule dynamics is regulated in peripheral and central-like axons in vitro. Using the microtubule plus-tip end-binding protein 3 (EB3) to track growing microtubules plus-tips (*Stepanova et al., 2003*), we observed that while both DRG axons had microtubule plus-end-out polarity (*Figure 3—video 2*), the central-like axon had a greater density of growing microtubules (*Figure 3C and D*), and a slower microtubule growth rate (*Figure 3E*). Notably, microtubule polymerization asymmetry was established early in DRG development, as it was already present in bipolar DRG neurons (*Figure 3C*).

The observation of microtubule asymmetries in DRG axons in vitro led us to investigate whether similar asymmetries exist in vivo. For that, we used Thy1-EB3-eGFP transgenic mice (*Kleele et al., 2014*), which express EB3-eGFP in neurons. DRG explants, including both the DRG peripheral nerve (containing peripheral axons) and the dorsal root (containing central axons) connected to the DRG

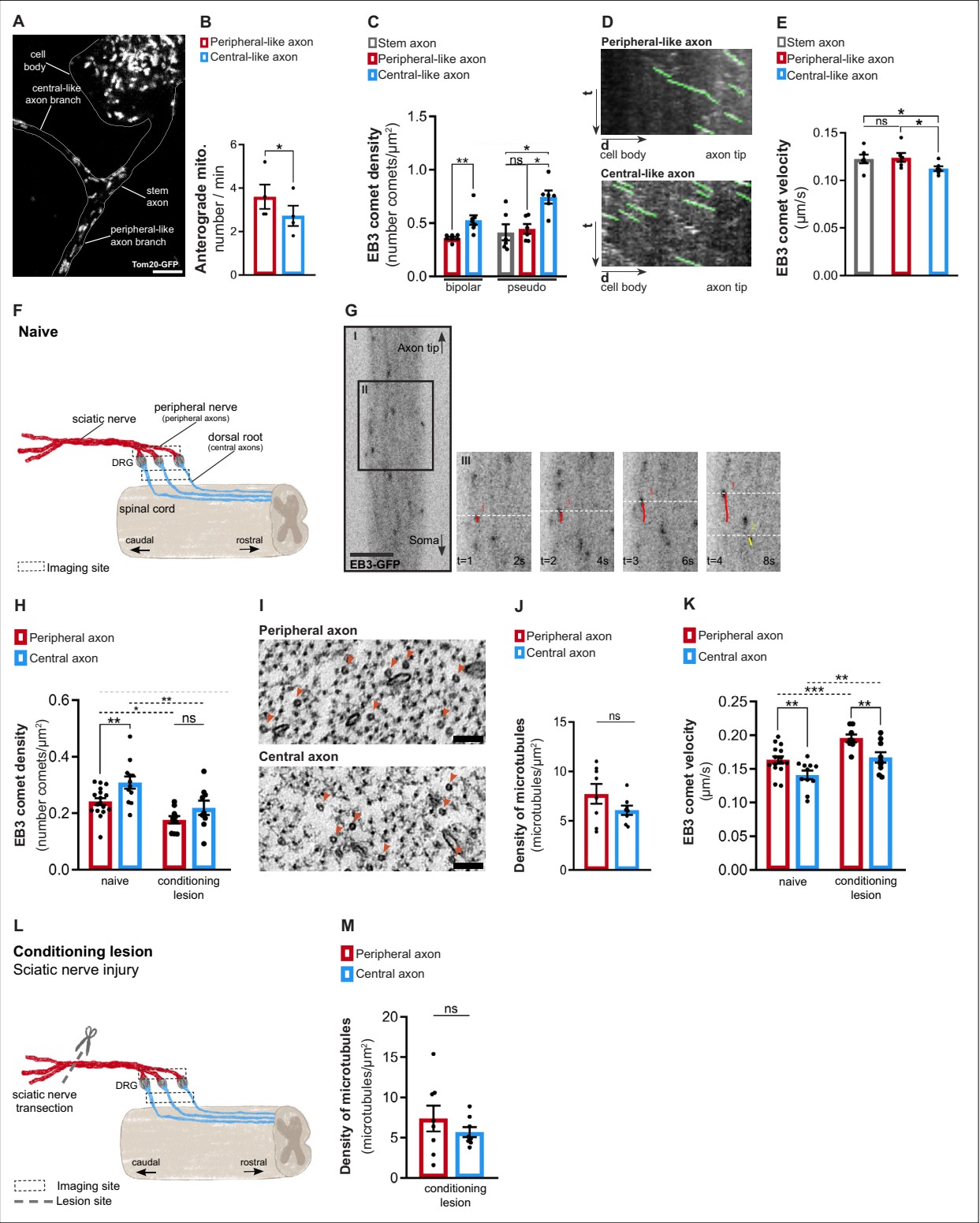

**Figure 3.** Dorsal root ganglion (DRG) axons display asymmetric microtubule polymerization, attenuated by a conditioning lesion. (**A**) In vitro pseudo-unipolar DRG neuron transduced with a Tom20-GFP lentivirus, labeling mitochondria. Scale bar, 5 µm. (**B**) Quantification of the anterograde mitochondria flux (n = 4 independent experiments, five DRGs/experiment; paired *t*-test, *p=0.0143). (**C**) EB3-GFP comet density in in vitro DRG axons (n = 6–7 independent experiments, 5–10 neurons/experiment; paired *t*-test in bipolar axons, **p=0.0038; repeated measures [RM] one-way ANOVA in pseudo-unipolar axons, stem-central *p=0.0221, peripheral-central *p=0.0171). (**D**) Kymographs of in vitro pseudo-unipolar DRG axons. (**E**) EB3-GFP comet velocity in in vitro pseudo-unipolar DRG axons (n = 6 independent experiments, 5–10 neurons/experiment; RM one-way ANOVA, stem-central

*Figure 3 continued on next page*

*Figure 3 continued*

*p=0.0443, peripheral-central *p=0.0183). (**F**) Representation of naive DRG neurons connected to the peripheral nerve (containing peripheral axons) and dorsal root (containing central axons). The dashed squares indicates the imaging locations. (**G**) Live imaging of DRG axons from Thy1-EB3-eGFP mice. Scale bar, 5 μm. (**H**) EB3-GFP comet density in DRG explants from naive mice (n = 12–17 animals; 3–6 axons/animal, **p=0.0037) and mice with a peripheral conditioning lesion (CL) (n = 9–10 animals, 3–5 axons/animal, p=0.1423). Two-way ANOVA; peripheral naive-peripheral CL, *p=0.0276; central naive central CL, **p=0.0026. (**I**) High-magnification electron microscopy images within individual naive DRG axons, depicting axonal microtubules (red arrowheads). Scale bar, 100 nm. (**J**) Total density of microtubules in naive DRG axons (n = 8 animals, 5–10 axons/animal; paired *t*-test, p=0.2299). (**K**) EB3-GFP comet velocity in DRG explants from naive mice (n = 11–15 animals, 3–6 axons/animal, **p=0.0048) and mice with peripheral CL (n = 8–9 animals, 3–5 axons/animal, **p=0.0035). Two-way ANOVA, peripheral naive-peripheral CL, ***p=0.0003; central naive-central CL, **p=0.0038. (**L**) Representation of a sciatic nerve injury to DRG neurons (conditioning lesion). The dashed square indicates the imaging location, while the dashed line and scissor marks the lesion site. (**M**) Total density of axonal microtubules in DRG peripheral and central axons after peripheral CL (n = 8 animals, five axons/animal; paired *t*-test, p=0.4624). Data are represented as mean ± SEM.

The online version of this article includes the following video(s) for figure 3:

**Figure 3—video 1.** Mitochondria transport in in vitro dorsal root ganglion (DRG) neurons.

https://elifesciences.org/articles/104069/figures#fig3video1

**Figure 3—video 2.** Microtubule dynamics in in vitro dorsal root ganglion (DRG) neurons.

https://elifesciences.org/articles/104069/figures#fig3video2

**Figure 3—video 3.** Microtubule dynamics in in vivo dorsal root ganglion (DRG) neurons.

https://elifesciences.org/articles/104069/figures#fig3video3

ganglia (containing DRG cell bodies) were collected and immediately imaged for up to 1 hr, proximal to the DRG ganglia (*Figure 3F*, dashed box). In both central and peripheral DRG axons, most microtubules moved with plus-end-out polarity (94% and 95%, respectively) (*Figure 3G*, *Figure 3—video 3*), as observed in vitro (*Figure 3—video 2*). Also replicating our in vitro observations (*Figure 3C*), central DRG axons exhibited an increased density of growing microtubules (*Figure 3H*, naive), which was unrelated to differences in total microtubule density (*Figure 3I and J*). Interestingly, and again replicating in vitro findings (*Figure 3E*), although having a higher density of growing microtubules, these grew at slower growth rates in central DRG axons (*Figure 3K*, naive). In summary, our findings indicate that although both axons originate from the same stem axon and share the same polarity, each DRG axon has distinct regulatory mechanisms governing its microtubule dynamics.

We next explored if this physiological asymmetry in microtubule polymerization is regulated after a conditioning lesion, where a global increase in axonal transport sustains central axon regeneration (*Mar et al., 2014b*). The shaft of peripheral and central DRG axons was imaged on explants 1.0–1.5 cm from the lesion site, 1-week post-lesion (*Figure 3L*, dashed box). Uninjured DRG explants were used as controls. A conditioning lesion decreased the density of growing microtubules not only in the peripheral DRG axon but this effect extended to the central DRG axon (*Figure 3H*, conditioning lesion), abolishing central-peripheral asymmetry. This was unrelated to differences in total microtubule density, as determined by electron microscopy (*Figure 3M*). In addition to decreasing the density of growing microtubules, a conditioning lesion increased the velocity of microtubule polymerization both in peripheral and central DRG axons (*Figure 3K*, conditioning lesion). Taken together, our findings show that injury to the peripheral DRG axon regulates microtubule dynamics in the shaft of both DRG axons, decreasing the density of growing microtubule plus-tips in central axons.

## DRG axons have a distinctive MAP signature that adapts upon conditioning lesion

To understand the mechanisms underlying asymmetry in microtubule dynamics in DRG axons, and its modulation by conditioning lesion, we analyzed tubulin post-translational modifications (PTMs) (*Janke and Magiera, 2020*) and microtubule-associated proteins (MAPs) (*Bodakuntla et al., 2019*) levels in DRG explants. We specifically selected the microtubule severing enzymes spastin and katanin as these enhance the formation of new microtubule plus-ends (*Kuo and Howard, 2021*). Whereas tubulin PTM levels were identical in both DRG peripheral nerve and dorsal root (*Figure 4—figure supplement 1*), the microtubule severases spastin and katanin were markedly increased in DRG central axons (*Figure 4B, C and, F*). In the case of spastin, the lack of specific antibodies for immunofluorescence precluded tissue analysis. Recently, the microtubule regulators CRMP5 (*Ji et al., 2018*; *Jin et al., 2022*) and tau (*Kadavath et al., 2015*; *Panda et al., 1995*; *Siahaan et al.,*

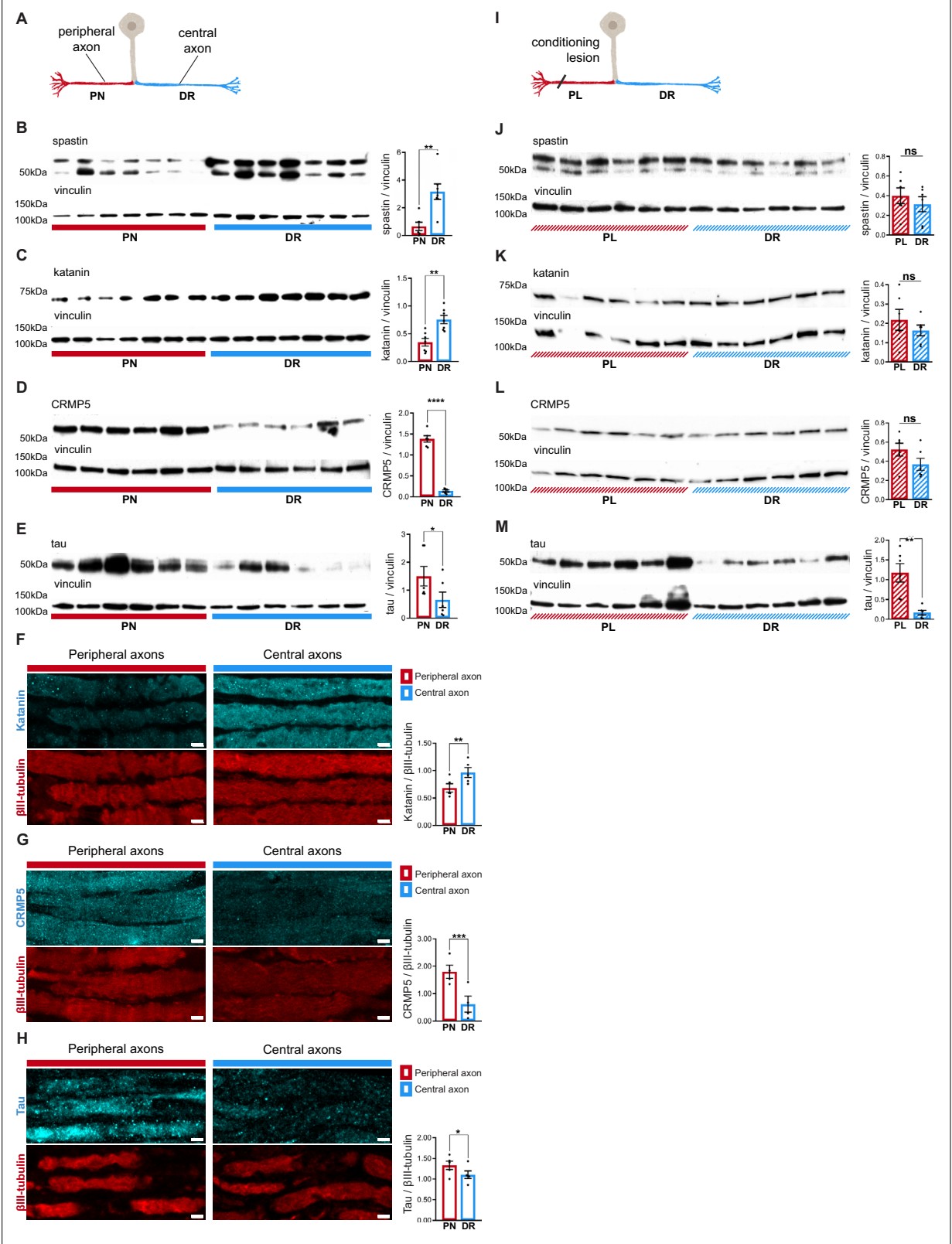

**Figure 4.** Dorsal root ganglion (DRG) axons have a distinctive microtubule-associated protein (MAP) signature that adapts upon conditioning lesion. (**A**) Representation of a naive DRG neuron. Peripheral axons within the peripheral nerve (PN) are depicted in red and central axons within the dorsal root (DR) in blue. (**B–E**) Western blot analysis of the PN and DR (n = 5–7 animals) and respective quantification of (**B**) spastin (unpaired *t*-test, **p=0.0032), (**C**) katanin (paired *t*-test, **p=0.0063), (**D**) CRMP5 (unpaired *t*-test, ****p<0.0001), and (**E**) tau (Wilcoxon test, *p=0.0313) levels.

*Figure 4 continued on next page*

*Figure 4 continued*

(**F–H**) Immunofluorescence of (**F**) katanin and βIII-tubulin, (**G**) CRMP5 and βIII-tubulin, and (**H**) tau and βIII-tubulin in DRG axons (n = 4–5 animals) and respective quantification (katanin, paired *t*-test, \*\*p=0.0021; CRMP5, paired *t*-test, \*\*\*p=0.009; tau, paired *t*-test, \*p=0.0114). Scale bar, 5 µm. (**I**) Representation of a DRG neuron with a priming peripheral lesion (peripheral lesion [PL]). (**J–M**) Western blot of DRG PN and DR following PL (n = 6 animals) and respective quantification (n = 6 animals; paired *t*-test) showing (**J**) spastin (p=0.4085), (**K**) katanin (p=0.2661), (**L**) CRMP5 (p=0.1326), and (**M**) tau (\*\*p=0.0052) levels. Data are represented as mean ± SEM.

The online version of this article includes the following source data and figure supplement(s) for figure 4:

**Source data 1.** Original MAP western blots seen in *Figure 4B–E and J–M*, indicating the relevant bands.

**Source data 2.** Original files for MAP western blots displayed in *Figure 4B–E and J–M*.

**Figure supplement 1.** Peripheral and central dorsal root ganglion (DRG) axons show similar levels of tubulin post-translational modifications.

**Figure supplement 1—source data 1.** Original PTM western blots for *Figure 4—figure supplement 1*, indicating the relevant bands.

**Figure supplement 1—source data 2.** Original files for the PTM western blots displayed in *Figure 4—figure supplement 1*.

**Figure supplement 2.** Potential sorting mechanism at the dorsal root ganglion (DRG) T-junction.

*2019*; *Tan et al., 2019*) were shown to inhibit the action of spastin and katanin by blocking their access to the microtubule lattice. Interestingly, DRG peripheral axons had increased levels of both microtubule regulators (*Figure 4D–E and G–H*), which by restricting severase access to microtubules may contribute to their decreased density of microtubule growing ends (*Figure 3H*, naive). Of note, CRMP5, similarly to CRMP2 (*Fukata et al., 2002*; *Hou, 2020*), may further support microtubule polymerization, enhancing microtubule growth rates in DRG peripheral axons (*Figure 3K*, naive). Hence, our data reveals that each DRG axon has a highly specific MAP signature that may support asymmetric microtubule polymerization.

Since microtubule dynamics adapts following conditioning lesion (*Figure 3H and K*), we analyzed whether it modulates the DRG MAP signature (*Figure 4I*). Interestingly, upon conditioning lesion, the uneven distribution of spastin, katanin, and CRMP5 in naive DRG (*Figure 4B–D*) was lost (*Figure 4J–L*). Importantly, the persistent asymmetry in tau distribution (*Figure 4E and M*) suggests that this MAP might be specifically related to the establishment of the structural differences between peripheral and central DRG axons.

The distinct MAP signatures of both DRG axons may arise from asymmetric axonal transport of mRNAs or proteins and/or differential local protein translation. However, as the density of *Spastin* and *Dpysl5* (CRMP5 mouse gene) transcripts in DRG axons was similar (*Figure 4—figure supplement 2A–C*), differential mRNA transport and local translation probably do not account for asymmetric distribution of these proteins. An alternative mechanism to establish asymmetry of DRG axons may rely on distinct microtubule streams entering peripheral and central axons. This is supported not only by early structural data (*Ha, 1970*), but also by our current in vitro model showing that microtubule asymmetry in DRG axons is established early in development. If these distinct streams exhibit different properties, they could guide specific cargoes to either central or peripheral axons. To investigate the arrangement of microtubules at the DRG T-junction, we analyzed in vitro DRG pseudo-unipolar neurons expressing EB3-GFP (*Figure 4—figure supplement 2D*) that allowed to assess whether microtubule polymerization is arrested (*Figure 4—figure supplement 2E*) or if a continuum of polymerization exists between the stem axon after the DRG bifurcation (*Figure 4—figure supplement 2F*). In peripheral-like axons, similar numbers of EB3 comets stopped or crossed past the bifurcation point (*Figure 4—figure supplement 2G*). Conversely, in DRG central-like axons, a ratio of 3:1 microtubules polymerized beyond the bifurcation point, supporting a higher microtubule continuum from the stem to the central-like axon (*Figure 4—figure supplement 2G*). As specific tubulin PTMs can favor the binding of particular motors (*Iwanski and Kapitein, 2023*), we investigated whether microtubules entering each DRG neuron axon at the DRG T-junction had different PTMs. While the levels of polyglutamylated and acetylated tubulin (*Figure 4—figure supplement 2H–K*) were the same in the two DRG axon compartments, Δ2 tubulin was the single PTM that was unevenly distributed, being increased in central-like axons (*Figure 4—figure supplement 2L–M*). Overall, it is possible that at the DRG T-junction, distinct PTMs and microtubule polymerization patterns may serve as mechanisms for selectively directing cargoes into specific DRG axons.

## Spastin deficiency is sufficient to abolish the asymmetry of DRG axons

To demonstrate the importance of the specific MAP signature of DRG axons, we investigated whether the absence of one of the signature proteins, spastin, was sufficient to affect DRG molecular asymmetries. In vitro, in wild-type and *Spastin* knockout (*Brill et al., 2016*) DRG neurons, the lack of spastin abolished axonal microtubule polymerization asymmetry (*Figure 5A and B*). We then crossed *Spastin* knockout mice with the Thy1-EB3-eGFP mouse line to test if this effect was also seen in vivo. Live imaging of DRG explants from *Spastin* knockout × Thy1-EB3-eGFP mice showed that the absence of this enzyme was sufficient to abolish the asymmetry of microtubule polymerization in DRG axons, both in terms of EB3 comet density (*Figure 5C and D*) and velocity (*Figure 5E and F*), while total microtubule density was preserved (*Figure 5—figure supplement 1A*). Together, our results show that interfering with a single protein of the DRG MAP signature is sufficient to disrupt the axonal asymmetry of microtubule polymerization.

Since spastin deficiency disrupted microtubule polymerization asymmetry of DRG axons, we then analyzed if it was sufficient to disrupt the conditioning lesion effect in vivo. At the time point at which lesions were performed, uninjured wild-type and *Spastin* knockout animals had a similar density of dorsal column tract (*Figure 5—figure supplement 1B*) and sciatic nerve axons (*Figure 5—figure supplement 1C and D*), ruling out primary defects of these tracts. For conditioning lesion, mice of both genotypes were subjected to sciatic nerve transection 1 week prior to spinal cord dorsal hemisection (*Figure 5G, I, and J*); wild-type animals with spinal cord injury alone were used as controls (*Figure 5H*). Regeneration was assessed 6 weeks post-spinal cord dorsal hemisection. In wild-type mice, conditioning lesion led to a substantial increase in the number of cholera toxin-positive regenerating DRG central axons (*Figure 5I and K*). In contrast, in *Spastin* knockout mice, the conditioning lesion effect severely reduced the number of central DRG axons being able to grow in the inhibitory environment of the lesion site (*Figure 5J and K*). As a generalized *Spastin* knockout mouse was used, to further support a neuron-specific effect, adult wild-type and *Spastin* knockout DRG neuron monocultures were grown on aggrecan to mimic the proteoglycan inhibitory environment formed upon spinal cord injury. Under these conditions, *Spastin* knockout DRG neurons extended 3.3-fold less primary neurites than wild-type DRG neurons (*Figure 5L and M*), replicating our in vivo findings. This indicates that loss of spastin in DRG neurons impairs axon growth under inhibitory conditions. In summary, interfering with the DRG MAP signature disrupts the asymmetry in microtubule polymerization of DRG axons as well as their ability to mount the conditioning effect.

## Discussion

DRG neurons are a striking example of polarity. An intriguing question concerning DRG biology is how peripheral and central axons linked to the same stem axon display different properties. One major setback in the field was the lack of an in vitro system recapitulating DRG development, pseudo-unipolarization, and establishment of polarity. Here, we developed an in vitro model where DRG neurons pseudo-unipolarize and mirror in vivo asymmetries, comprising those found in axon diameter, regenerative capacity, axonal transport, and microtubule dynamics (*Figure 6A*). This model serves as a valuable resource to understand DRG neuron biology (including stem axon formation and possible sorting at the DRG T-junction), mechanisms driving axon regeneration asymmetries, and DRG-related diseases. Regarding DRG stem axon formation, our findings support that it may be driven by cell body bulging rather than fusion of the two neurites, aligning with previous studies (*Matsuda et al., 2000*). Concerning potential asymmetries in protein transport and microtubule dynamics at the DRG T-junction, while investigating these in vivo is challenging due to the length of the stem axon and the difficulty of imaging the T-junction, they can now be effectively studied using our in vitro system. Future studies should further address neuron-intrinsic mechanisms driving pseudo-unipolarization (such as the role of cytoskeletal dynamics, and of Golgi and nuclear positioning), as well as the influence of extrinsic factors, namely the identify of specific DRG glial cells supporting pseudo-unipolarization. While the current model focuses on features shared across DRG subtypes, such as pseudo-unipolarization and the higher regenerative capacity of peripheral branches, whether it favors specific DRG subtypes should be further explored. In vitro models resourcing to DRG explants and the development of compartmentalized systems with peripheral and central targets could provide additional insights into pseudo-unipolarization, regenerative capacity, and target re-innervation. However, although previous

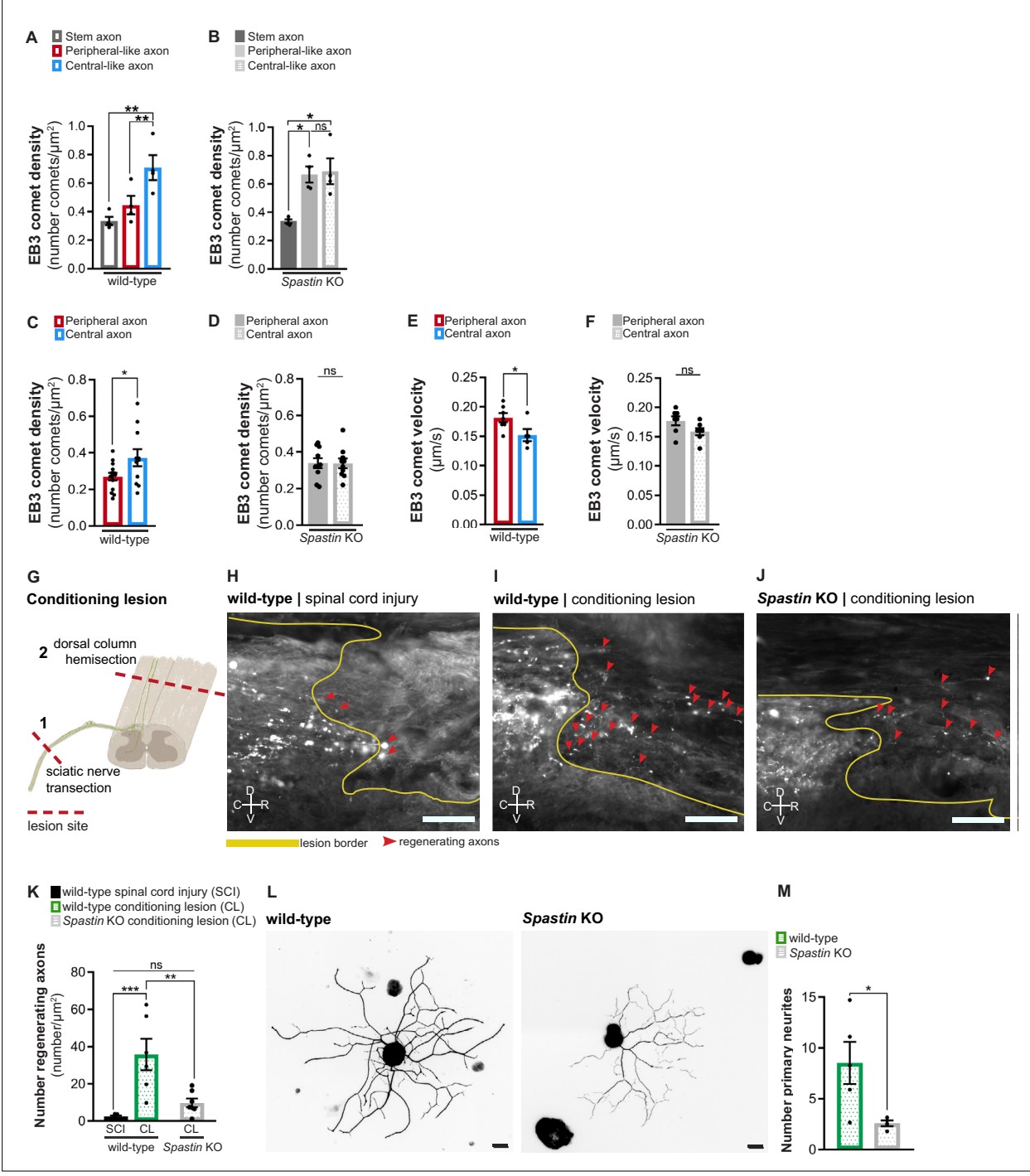

**Figure 5.** Spastin deficiency is sufficient to abolish the asymmetry of dorsal root ganglion (DRG) axons. (**A**) Density of EB3-GFP comets in wild-type DRG axons in vitro (n = 4 independent experiments, 10 cells/experiment; repeated measures [RM] one-way ANOVA, stem-peripheral p=0.0536, stem-central **p=0.0082, peripheral-central **p=0.0027). (**B**) Density of EB3-GFP comets in *Spastin* knockout DRG axons in vitro (n = 4 independent experiments, 10 cells/experiment; RM one-way ANOVA, stem-peripheral *p=0.0168, stem-central *p=0.0250, peripheral-central p=0.8762). (**C, D**) EB3-eGFP comet density in (**C**) wild-type and (**D**) *Spastin* knockout mice (n = 10–14 animals; three axons/animal; unpaired *t*-test; wild-type, *p=0.0388; knockout, p=0.9792). (**E, F**) EB3-eGFP comet velocity in (**E**) wild-type and (**F**) *Spastin* knockout mice (n = 5–7 animals; three axons/animal; unpaired *t*-test; wild-type, *p=0.0405; knockout, p=0.0823). (**G**) Representation of the conditioning lesion (CL). A dorsal column hemisection is preceded by a sciatic nerve transection 1 week before. Lesion sites are indicated with dashed red lines and DRG axons in green. (**H–J**) Longitudinal spinal cord sections of (**H**) wild-type mice with spinal cord lesion or (**I**) CL and (**J**) *Spastin* knockout mice with CL. Dorsal column tract axons were traced with cholera toxin-B (white). The lesion border is highlighted by a yellow line. Regenerating axons are highlighted by red arrowheads. C, caudal; R, rostral; D, dorsal; V, ventral. Scale bar,

*Figure 5 continued on next page*

*Figure 5 continued*

100 μm. (**K**) Number of regenerating axons in wild-type mice with spinal cord injury (n = 5 animals) and CL (n = 6 animals), and *Spastin* knockout with CL (n = 7 animals); six sections per animal. One-way ANOVA; wild-type SCI-CL, ***p=0.0005; wild-type-knockout CL, **p=0.0020; wild-type SCI-knockout CL, p=0.3335. (**L**) Representative in vitro wild-type and *Spastin* knockout adult DRG neurons labeled with βIII-tubulin. Scale bar, 30 μm. (**M**) Quantification of the number of primary neurites in adult wild-type and *Spastin* knockout DRG neurons. n = 4–5 independent experiments for wild-type and *Spastin* knockout; unpaired *t*-test; *p=0.0205. Data are represented as mean ± SEM.

The online version of this article includes the following figure supplement(s) for figure 5:

**Figure supplement 1.** Signs of axon degeneration are not observed in 15-week-old *Spastin* knockout animals.

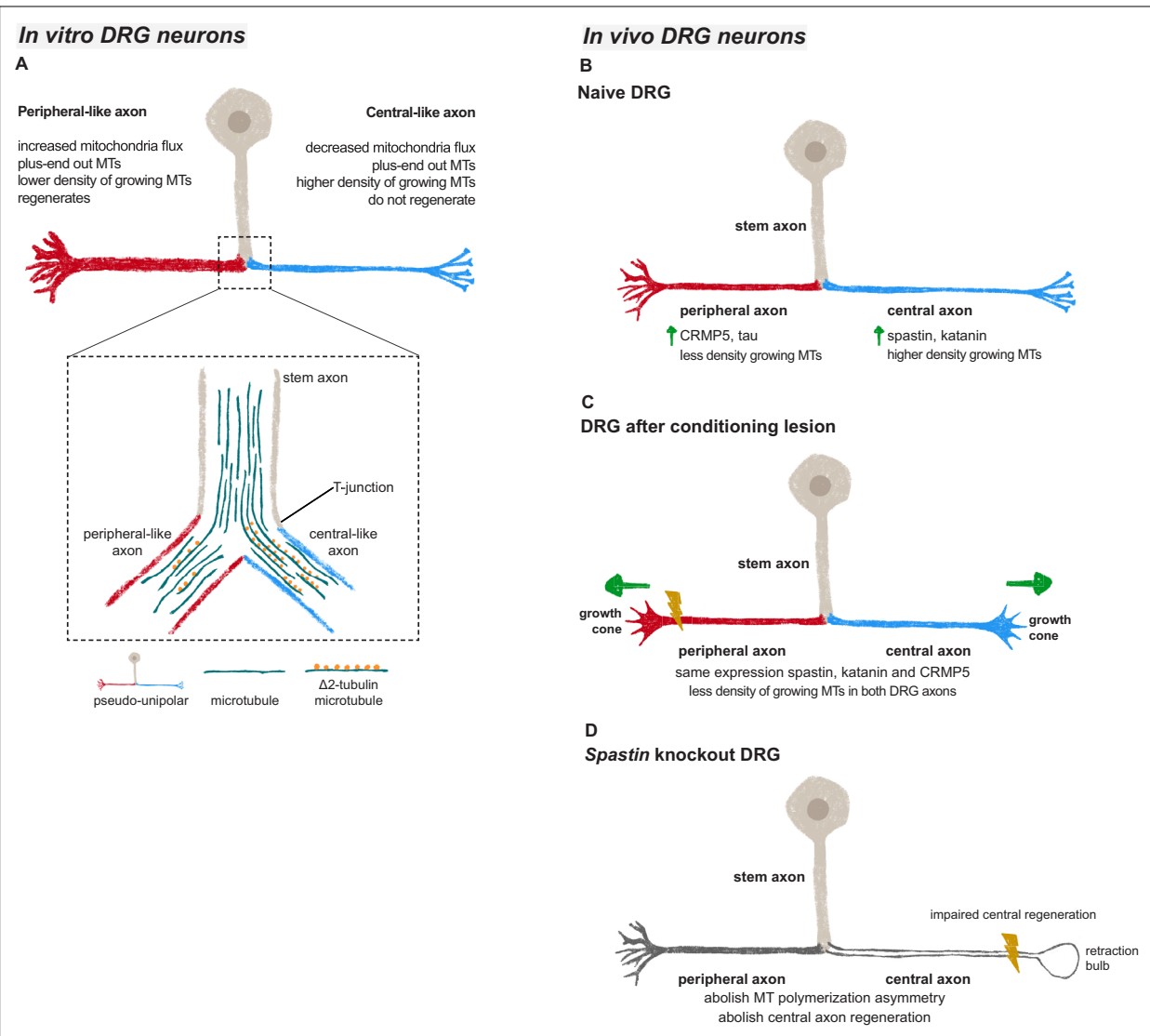

**Figure 6.** Microtubules in dorsal root ganglion (DRG) axons are distinctively regulated both in vivo and in vitro. (**A**) In vitro DRG cultures replicate in vivo asymmetry in microtubule polymerization (established already at the bipolar stage) and reveal a potential cargo filtering mechanism operating at the DRG T-junction, where a higher continuum of microtubules from the stem axon to the central-like axonal branch is found, together with increased levels of Δ2 tubulin in the central-like axonal branch. (**B**) Under physiological conditions, peripheral DRG axons show a decreased density of growing microtubules when compared to central axons due to an asymmetric MAP signature. (**C**) After a peripheral conditioning lesion, the DRG axon MAP signature is remodeled, leading to a decrease in the density of growing microtubules and regeneration of both peripheral and central axons. (**D**) Perturbing the DRG MAP signature by knocking out *Spastin* abolishes DRG axon asymmetry and central axon regeneration following conditioning lesion.

compartmentalized DRG cultures have been reported (*Giorgi et al., 2023*; *Vikman et al., 2001*), these have not demonstrated pseudo-unipolarization.

We found that the asymmetric microtubule dynamics of peripheral and central DRG axons, which is observed both in vitro and in vivo, is supported by a distinct MAP signature that promotes a higher density of growing microtubules in central axons (*Figure 6B*). Despite scattered reports in the literature, markers of distinct DRG axons remained unidentified (*García-Añoveros et al., 2001*; *Katano et al., 2006*; *Papasozomenos et al., 1985*) and early studies suggested that similar proteins are transported to both DRG axons (*Bisby, 1981*; *Perry and Wilson, 1981*; *Stone and Wilson, 1979*). Our data shows, however, that the content of specific microtubule modulators including severases and the microtubule regulators CRMP5 and tau is axon-specific in DRG neurons. Other MAPs, such as MAP2, show however no differences between the two DRG axons. The mechanisms underlying this differential protein distribution remain unclear. One hypothesis is that mRNAs or proteins are asymmetrically transported at the DRG T-junction. While our data show no significant differences in the density of *Spastin* and *Dpysl5* mRNA between both DRG axons, the possibility of asymmetric protein transport cannot be ruled out. Recently, a pre-axonal cargo filtering zone mediated by MAP2 was identified in DRG neurons (*Gumy et al., 2017*). Although this may partially establish two separate compartments – the cell body and axons – how cargo filtering takes place at the DRG T-junction remains unknown. In multipolar neurons, differential cargo sorting to dendrites or axons is achieved through the axon initial segment (AIS) (*Song et al., 2009*) and by the distinct microtubule polarity in axons and dendrites (*Zheng et al., 2008*). Similar mechanisms are however difficult to envisage in DRG neurons. Not only microtubules in both DRG axons are oriented plus-ends out, but also the AIS of DRG neurons is located in the stem axon proximal to the DRG cell body (*Nascimento et al., 2022*). An alternative mechanism to establish asymmetry of DRG axons may rely on distinct microtubule streams entering peripheral and central axons as supported by early structural data (*Ha, 1970*). In fact, our data suggest that there is a preferential continuous microtubule polymerization from the DRG stem axon into the central-like axon that is accompanied by increased Δ2 tubulin levels. This differential microtubule organization may serve as a mechanism for cargo filtering between DRG axons. In chemotherapy-induced peripheral neuropathy, the accumulation of Δ2 tubulin in DRGs and peripheral nerves disrupts mitochondrial motility (*Pero et al., 2021*). Accordingly, at the DRG T-junction the anterograde transport of mitochondria is biased toward the DRG peripheral-like axon, where Δ2 tubulin levels are reduced. It is thus possible that Δ2 tubulin interferes with binding of molecular motors, contributing to establish a cargo filtering mechanism at the DRG T-junction. In the future, it would be valuable to investigate whether microtubules at the DRG T-junction intermix or segregate as separate pools as they extend from the cell body towards the bifurcation.

DRG neurons are a powerful model to study axon regeneration given the distinct regenerative capacity of their axons, and the fact that a peripheral conditioning lesion enables regeneration of central axons (*Neumann and Woolf, 1999*; *Richardson and Verge, 1987*). We previously demonstrated that a conditioning lesion elicits a global increase in axonal transport that extends to the central DRG axon, allowing rapid and sustained support of central axon regrowth (*Mar et al., 2014b*). Here we show that a conditioning lesion also induces alterations in the DRG microtubule cytoskeleton, decreasing the density of growing microtubules through an adaptive response of the MAP signature of DRG axons (*Figure 6C*). This decrease in microtubule dynamics extends away from the injury site, propagating to the distant axon shaft of both DRG branches. The study of changes in axonal microtubule dynamics has been mostly restricted to local effects in the growth cone. In this compartment, low doses of the microtubule-stabilizing drug taxol reduce the percentage of retraction bulbs after central axon injury (*Ertürk et al., 2007*; *Hellal et al., 2011*). Microtubule stabilization is also crucial to sustain efficient axonal transport, with taxol-stabilized microtubules enhancing kinesin binding (*Dunn et al., 2008*; *Kapitein et al., 2010*). Likewise, the reduction in growing microtubule density following a conditioning lesion may contribute to microtubule stabilization, supporting axonal transport and regeneration. Similarly to the growth cone where a delicate balance between microtubule stabilization and dynamics is necessary, a balanced regulation of microtubule dynamics in the axon shaft may also be required, as excessive stabilization may be detrimental (*Gornstein and Schwarz, 2017*; *Pero et al., 2021*).

Maintaining the MAP signature is crucial for preserving the physiology and regenerative capacity of DRG neurons. Our data strongly underscore the significance of this signature, showing that the loss

of spastin disrupts microtubule asymmetry in DRG neurons and results in reduced central axon regeneration (*Figure 6D*). Examining the effects of depleting other MAPs, such as katanin and CRMP5, which lose their asymmetry following conditioning lesion, could offer additional valuable insights into their roles in maintaining DRG axon asymmetry and supporting axon regeneration. Notably, katanin shares similar severing mechanisms with spastin (*Roll-Mecak and Vale, 2008*; *Zehr et al., 2017*) and is partially regulated in a comparable manner. Both proteins undergo biphasic regulation through tubulin glutamylation (*Lacroix et al., 2010*; *Szczesna et al., 2022*; *Valenstein and Roll-Mecak, 2016*) and their severing activity is inhibited by tau condensates (*Siahaan et al., 2019*; *Tan et al., 2019*). Moreover, katanin plays a role in axon outgrowth (*Ahmad et al., 1999*; *Chen et al., 2014*), with RhoA negatively modulating neurite outgrowth via regulation of spastin and katanin (*Tan et al., 2020*). Despite not being functionally redundant, it is possible that katanin deficiency may impact DRG microtubule asymmetry and regenerative capacity similarly to spastin. In summary, our data unveil that axon-specific microtubule regulation drives asymmetric regeneration of sensory neuron axons. We further provide a bona fide model of DRG pseudo-unipolarization and establishment of axonal asymmetry that will contribute to push forward our understanding of sensory neuron biology and of axon growth and regeneration.

## Materials and methods

### Culture of pseudo-unipolar DRG neurons

DRG cultures from Wistar rats (embryonic day 16) and *Spastin* knockout mice and wild-type littermates (embryonic day 13) were done as previously described (*Nascimento et al., 2022*). These developmental stages in rats and mice yield DRG neuron cultures with similar percentages of pseudo-unipolarization. Briefly, DRGs were isolated and mechanically dissociated, digested with trypsin 0.05% EDTA (#25300062, Gibco) for 1 hr at 37°C, centrifuged at $200 \times g$, and mechanically dissociated with a 1000 µl micropipette, in the final culture medium (Neurobasal medium supplemented with 2% B27 [#17504044, Gibco], 2 mM L-glutamine, 1% penicillin–streptomycin, and 50 ng/ml of nerve growth factor [#N-100, Millipore]). Cells were then seeded either at 27,000 (for analyses of microtubule polymerization), 7000 (for studies using laser ablation and stem axon formation), or at 3000 (for immunocytochemistry) cells/well in glass-bottom 4-well chambered coverslips (#80427, Ibidi) (for live imaging) or in 24-well plates (for immunocytochemistry), previously precoated with 20 µg/ml poly-L-lysine and 5 µg/ml laminin. Cultures were maintained at 37°C in a 5% $CO_2$ incubator and the medium was partially changed once a week for 21 days, a time point at which pseudo-unipolarization occurs in approximately 43 ± 3% of cultured DRG neurons. Pseudo-unipolar DRG neurons were identified based on the following criteria: the presence of a clearly discernible stem axon measuring at least 10 µm in length, with the proximal region of the bifurcating axons devoid of crossing neurites within a distance of 20–30 µm, ensuring an unobstructed area for analysis. Peripheral-like and central-like axons were distinguished by consistently measuring their diameters across all experiments.

### Analysis of stem axon formation in DRG neuron cultures

To investigate the in vitro formation of the stem axon in DRG neuron development, we transduced these cells with an ultra-purified recombinant AAVPHP.S-CMV-eGFP (#VB010000-9394npt, VectorBuilder) virus at DIV 9 at a concentration of $1.12 \times 10^{10}$ cfu/ 7000 cells. We live imaged the cells for seven consecutive days from DIV 14 to 21. Approximately 30 bipolar and bell-shaped DRG neurons per well, exhibiting a low to moderate eGFP expression, were selected. Imaging was performed using a widefield inverted microscope, Nikon ECLIPSE Ti (Nikon, Japan), equipped with an Iris 9 sCMOS camera (Photometrics) and a CFI Plan Apochromat Lambda D ×40/0.95 DIC objective, and controlled through the NIS elements 5.4 software (Nikon). eGFP excitation was achieved using the Lumencor Spectra X Cyan 470/24 with LED % of 10, an exposure time of 200 ms, in combination with the following filters: excitation filter: FF01-387/485/559/649; dichroic: FF410/504/582/669-Di01; emission filter: FF01-440/521/607/700. Images were acquired every hour. Gray levels were inverted for improved visualization. Due to the random nature of viral infection and the potential light-induced toxicity associated with the longitudinal imaging setup, only a subset of the 30 selected neurons underwent pseudo-unipolarization in each experiment. To quantify the initial and final diameter and length of the stem axon, the Fiji/ImageJ segmented line tool was employed. To quantify the displacement of

the base of the stem axon and of the top of the DRG cell body, we used the initial position of the base of the DRG T-junction as a reference point. Using this reference point, we tracked the displacement of both the base of the stem axon and the top of the cell body using the MTrackJ plugin in ImageJ/ Fiji software. For DRG segmentation, we acquired pseudo-unipolar eGFP-positive DRG neurons at DIV14 using an inverted Leica SP8 single-point scanning confocal microscope (Leica Microsystems, Germany) and Leica Application Suite X software (version 3.5.7.23225, RRID:SCR_013673). All images were scanned unidirectionally at 400 Hz using the galvanometer-based imaging mode, with a z-step size of 0.2 µm. The final image resolution was 0.12 µm/pixel. Pseudo-unipolar DRG neurons were selected for segmentation using Imaris software (version 10.1.1). The Imaris Surfaces tool was applied to extract a 3D object from the DRG neuron outer contour using an intensity threshold to distinguish the cell from the background. We then used the Imaris Animation tool to create a video of the segmented DRG neuron.

## Axotomy of cultured pseudo-unipolar DRG neurons

To study the regenerative capacity of pseudo-unipolar DRG neurons in vitro, we performed a laser axotomy of large and small-diameter axons. At DIV 15, rat DRG neurons were transduced with an ultra-purified recombinant AAVPHP.S-CMV-eGFP (#VB010000-9394npt, VectorBuilder) virus at 1.12 × $10^{10}$ cfu/ 7000 cells. At DIV 21, neurons were live imaged using an inverted 3i Marianas Spinning Disk confocal system with an Yokogowa W1 scanhead+ illumination Uniformizer, a Prime 95b sCMOS camera, mounted on a fully motorized Nikon Ti2-E microscope, and using a CFI Plan Apo VC 60XC 1.2 water/silicone immersion objective. The system was also equipped with a 3i Vector (galvo-based laser positioning unit)+3i !Ablate unit with a 355 nm pulsed laser. DRG neurons were first pre-screened to select target axons to injure (100–200 µm distally from the bifurcation point of the stem axon) and the areas to be imaged; six pseudo-unipolar DRG neurons were selected per experiment. eGFP was excited using a 488 nm laser line with a laser power of 7%. Laser ablation was performed on both large and small diameter axons using a 355 nm laser line that creates accurate and precise injuries to single axons while minimizing damage to nearby cells (*Cengiz et al., 2012*). We limited the injuries to a laser power of 50% and restricted them to three pulses per axon. Injuries were performed at 100–200 µm from the stem axon to minimize disruptions to intracellular homeostasis, which are more pronounced in lesions closer to the cell body and may lead to cell death (*Cengiz et al., 2012*). To ensure clear visualization of the regeneration process, neurons were selected when no crossing neurites from other cells were found within 100–200 µm of the stem axon. Imaging conditions and images were acquired using the Slidebook software (version 2022, RRID:SCR_014423) with a pinhole size of 50 µm and a final image resolution of 0.18 µm/ pixel. All images were acquired every 4 min for 5 hr. Five independent experiments were performed to analyze axon retraction (length and duration) and axon regeneration (regenerative length and frequency of regenerating versus non-regenerating axonal branches). Only axons that grew throughout the 300 frames (5 hr) were considered regenerating axons. Gray levels were inverted to facilitate visualization.

## Live imaging of mitochondria transport in pseudo-unipolar DRG neuron cultures

DRG neurons from E16 Wistar rats were maintained in culture until DIV 21. Then, 40 hr before live imaging, cells were transduced with a lentivirus expressing synapsin-Tom20-GFP (#BLV-569a, Charité NeuroCure, University of Berlin) at 7.07 × $10^5$ cfu/7000 cells. Images were acquired using an inverted Leica SP8 single-point scanning confocal microscope (Leica Microsystems) and a Leica Application Suit X software (version 3.5.7.23225). All images were acquired every 2 s, for 63 frames, scanning unidirectionally at 400 Hz, using the galvanometer-based imaging mode, a line average of 3, and a digital zoom of 3×. Experiments were conducted at 37°C and 5% $CO_2$. All images had a final resolution of 0.12 µm/pixel. Only pseudo-unipolar DRG neurons with low-to-moderate expression levels of Tom20-GFP were selected. The flux of anterogradely moving mitochondria in the DRG large and thin axons was determined as the number of mitochondria crossing a vertical line (placed in the middle of the axon) per minute. Five pseudo-unipolar DRG neurons were quantified in each experiment.

## Live imaging of microtubule polymerization in pseudo-unipolar DRG neuron cultures

DRG neurons from E16 Wistar rats, and E13 *Spastin* knockout mice were maintained in culture for 21 days in vitro. Then, 40 hr before live imaging, cells were transduced with a lentivirus expressing CMV-EB3-GFP (from Franck Polleux) at $2.7 \times 10^4$ cfu/27.000 cells. Images were acquired using an inverted Leica SP8 single-point scanning confocal microscope and a Leica Application Suit X software. All images were acquired every 2 s, for 46 frames, scanning unidirectionally at 400 Hz, using the galvanometer-based imaging mode, a line average of 4, and a digital zoom of 3.25×. Experiments were conducted at 37°C and 5% $CO_2$. All images had a final resolution of 0.11 µm/pixel. Only bipolar and pseudo-unipolar DRG neurons with low-to-moderate expression levels of EB3-GFP were selected. The EB3-GFP comet velocity and density were quantified through kymographs using the Fiji/ImageJ KymoResliceWide plugin (*Katrukha, 2020*) or the KymoToolBox plugin (*Zala et al., 2013*). Gray levels were inverted to facilitate visualization. A segmented line was drawn over the DRG stem, large and thin axons, in the direction cell body-axon tip, and maximum intensity was extracted across the line width, when using the KymoResliceWide plugin. For EB3-GFP comets analyzed using the KymoResliceWide plugin, starting and end positions of the kymograph traces were defined using the Fiji/ImageJ Cell Counter plugin. The x-axis and y-axis coordinates of each trace were used to calculate the EB3-GFP comet velocity (distance, x-axis; time, y-axis). EB3-GFP comet density was calculated by dividing the total number of EB3-GFP comets counted within one kymograph by the area of the segmented line. For EB3-GFP comets analyzed using the KymoToolBox plugin, comet trajectories were manually identified using the segmented line tool. Approximately 5–10 bipolar and pseudo-unipolar DRG neurons were quantified in each experiment. The videos used to quantify EB3-GFP comet dynamics were used to quantify axon diameter in in vitro bipolar and pseudo-unipolar DRG neurons. For that, the diameter of the stem, large and thin axons was determined using the straight-line tool from Fiji/ImageJ to trace a perpendicular line connecting the two axon membrane borders. The videos of pseudo-unipolar DRG neurons where the stem, large, and thin axons were located on the same focal plane were used to quantify the total number of EB3-GFP comets that stop and cross the DRG T-junction. The number of EB3-comets that cross or stop at the DRG T-junction was traced using the composite panel obtained after analysis with the KymoToolBox plugin in a length of approximately 5 µm comprising the final part of the stem axon and the beginning of the large or thin axon.

## Animals

Animal experiments followed EU Directive 2010/63/EU and national Decree-law number 113-2013 approved by the i3S Ethical Committee and Portuguese Veterinarian Board. Unless otherwise stated, C57BL/6 adult mice at 8–11 weeks of age were used. Thy1-EB3-eGFP mouse line was used to study microtubule polymerization (*Kleele et al., 2014*). Constitutive *Spastin* knockout mice (*Brill et al., 2016*) and wild-type littermates were used and genotyped as detailed in *Brill et al., 2016*. Mouse strains and Wistar rats were bred at the i3S animal facility, maintained with ad libitum access to water/food, and kept on a 12 hr light/dark cycle under controlled temperature and humidity. Power analysis and previous experience from the lab guided sample size determination. Animals of both genders were used in the experiments.

## Animal surgeries

For sciatic nerve injury (peripheral injury), bilateral sciatic nerve transection was done in adult mice 2 mm proximal to the sciatic nerve trifurcation. Analgesia was provided to animals (3 mg buprenorphine/kg; administered twice a day). One week later, the mice were euthanized, and lumbar DRG L3, L4, and L5 explants were isolated. For live imaging experiments, both the dorsal root (containing central DRG axons) and the peripheral nerve (containing peripheral DRG axons) were imaged attached to the DRG ganglia, while for western blots, immunohistochemistry, electron microscopy, and RNAscope, the dorsal root, and the peripheral nerve were separated from the ganglia.

For central lesions, wild-type and *Spastin* knockout mice underwent spinal cord dorsal hemisection at the T8 level using a micro-feather ophthalmic scalpel (#72045-45, Delta Microscopes). For conditioning lesions, a bilateral complete sciatic nerve transection was performed first, followed a week later by spinal cord dorsal hemisection. Regeneration of central axons was evaluated after 6 weeks post-spinal cord injury. During this period, animals received analgesia, and bladder voiding

was supported twice daily via gentle abdominal compression. To trace regenerating dorsal column axons, 2 µl of 1% cholera toxin-B (List Biologicals, 103B) was injected into the left sciatic nerve with a 10 µl Hamilton syringe 4 days before euthanasia. Afterward, animals were perfused with 4% paraformaldehyde, and their spinal cords were post-fixed for 1 week at 4°C before being cryoprotected in 30% sucrose.

## Analysis of microtubule polymerization in DRG explants

DRG ganglia with attached dorsal root (containing central DRG axons) and peripheral nerve (containing peripheral DRG axons) were placed in a 35 mm imaging dish with a glass bottom (#81158, Ibidi). Live imaging recordings were performed in phenol-free neurobasal medium (#12348017, Gibco) supplemented with 2% B27, 2 mM L-glutamine, 1% penicillin–streptomycin, and 50 ng/ml of nerve growth factor. To study microtubule polymerization, imaging of DRG explants from Thy1-EB3-eGFP and *Spastin* knockout and wild-type adult mice was performed using an Andor Revolution XD (Andor Technology) inverted Olympus IX81 (Olympus, UK) spinning disk confocal microscope. An Olympus UPlanLSAPOpo ×100/1.40 Oil immersion objective with a Zeiss Immersol 518F was used. Data was acquired using IQ 2 software (Andor Technology, UK). Experiments were conducted in an environmental chamber at 37°C and 5% $CO_2$. Images were obtained every 2 s, for 80 frames, with an exposure time of 200 ms, z-range of 6 µm, z-step size of 2 µm, and a final digital resolution of 0.08 µm/pixel. For the analysis of microtubule polymerization in DRG explants, only axons with low to moderate expression of eGFP were selected. Four central and peripheral DRG axons in Thy1-EB3-eGFP mice and three central and peripheral DRG axons in *Spastin* knockout and wild-type mice were quantified per animal. The MTrackJ plugin (*Meijering et al., 2012*) of Fiji/ImageJ software (*Schindelin et al., 2012*; RRID:SCR_002285) was used. Gray levels were inverted to facilitate visualization. The velocity of 30 EB3-eGFP comets per axon was quantified during the first 40 frames of each video. To determine the density of polymerizing microtubules, the total number of EB3-eGFP comets contained in an ROI of 100 $µm^2$ during the first 15 frames of each video was quantified.

To analyze total microtubule density, DRG dorsal root and peripheral nerve were postfixed with 2% osmium tetroxide, dehydrated with ascending series of ethanol, and embedded in Epon. The samples were cut at a thickness of 50 nm and observed using a transmission electron microscope JEOL JEM-1400, at a 40K times magnification and a final resolution of 0.006 µm/pixel. Fiji/ImageJ Cell Counter plugin was used to quantify the total number of microtubules within each axon. Only DRG axons with a diameter between 6 and 11 µm were quantified. The density of microtubules was obtained by dividing the total number of microtubules by the axonal area. Up to 10 axons per DRG branch were quantified.

## Immunoblotting

Immunoblots were done on adult naive and sciatic nerve-injured (conditioning lesion) C57BL/6J mice. DRGs from the lumbar roots L3, L4, and L5, which together contribute to the sciatic nerve in this species, were isolated. To minimize contamination from motor axons of the ventral root, a ventral root rhizotomy was performed. Subsequently, the dorsal root (containing DRG central axons) and the peripheral nerve (containing DRG peripheral axons) immediately adjacent to the ganglia were isolated and prepared with ice-cold lysis buffer (1% Triton X-100, 0.1% SDS, 140 mM sodium chloride diluted in Tris-EDTA buffer pH 8.0, and containing protease inhibitors [#04693124001, Roche]). Protein levels were determined using the Detergent Compatible Protein Assay (#5000116, Bio-Rad). 1 µg of total protein were used to determine the expression levels of tubulin PTMs and 7.5 µg of total protein were used to determine the expression of MAPs. Proteins were loaded onto 4–20% acrylamide gels (#5671093, Bio-Rad). Gels were transferred to nitrocellulose membranes (#10600013, GE Healthcare Life Sciences), and the primary antibodies were probed overnight at 4°C in 5% bovine serum albumin. The following primary antibodies were used: rat anti-tyrosinated tubulin YL1/2 (1:5000, #ab6160, Abcam, RRID:AB_305328), mouse anti-acetylated tubulin (1:10,000, #T7451, Sigma-Aldrich, RRID:AB_609894), rabbit anti-Δ2 tubulin (1:2000, #AB3203, Millipore, RRID:AB_177351), rabbit anti-polyglutamylated tubulin (1:10,000, # AG-25B-0030-C050, Adipogen), mouse anti-spastin (1:500, #sc-81624, Santa Cruz, RRID:AB_2286628), rabbit anti-katanin p60 (1:500, #17560-1-AP, Proteintech, RRID:AB_10694670), mouse anti-tau (1:1000, #4019, Cell Signalling), rat anti-CRMP5 (1:200, #sc-58515, Santa Cruz, RRID:AB_782270), and rabbit anti-vinculin (3:10,000, #700062, Invitrogen).

HRP-conjugated secondary antibodies were diluted in 5% skim milk and incubated for 1 hr at room temperature. Membranes were either exposed to Fuji Medical X-Ray Film (#16195209, Fujifilm) and scanned using a Molecular Imager GS800 (Bio-Rad) or directly scanned using a ChemiDoc Imaging System (Bio-Rad). Scanned membranes were quantified using Image Lab software (Bio-Rad, RRID:SCR_014210).

## Immunofluorescence of peripheral and central DRG axons

L3, L4, and L5 mouse DRGs were dissected and fixed with 4% paraformaldehyde for 24 hr at 4°C, then transferred into cryoprotection solution and embedded in OCT. Longitudinal sections of 12 μm were prepared using a cryostat and stored at –80°C. The tissues were thawed for 30 min at 37°C and washed three times with PBS to remove all the OCT. Tissue permeabilization was done with 1% triton during 30 min for katanin and with methanol at room temperature for 20 min for CRMP5 and tau. Antigen retrieval was performed using Citrate pH 6 for tau and Tris-EDTA pH 9 for CRMP5 and katanin, for 10 min at approximately 80–90°C. Endogenous autofluorescence was blocked with ammonium chloride for 30 min, followed by IgG blocking with M.O.M. blocking reagent (Vecto Laboratories, # MKB-2213) for 1 hr at room temperature. Tissues were further blocked with 5% normal donkey serum for 30 min. Primary antibodies, namely rabbit anti-KATNA1 (1:500, Proteintech, #17560-1-AP), rabbit anti-CRMP5 (1:200, Abcam, #ab36203), mouse anti-tau (1:100, Cell Signalling, #4019), rabbit anti-βIII-tubulin (1:200, Synaptic Systems, #302302), and mouse anti-βIII-tubulin (1:500, Promega, #G7121) were incubated overnight at 4°C in blocking buffer. The next day, secondary antibodies Alexa Fluor 488 anti-mouse (1:250, Jackson ImmunoResearch, #715-545-15), Alexa Fluor 488 anti-rabbit (1:250, Jackson ImmunoResearch, #711-545-152), Alexa Fluor 594 anti-rabbit (1:250, Jackson ImmunoResearch, #711-585-152), and Alexa Fluor 594 anti-mouse (1:250, Jackson ImmunoResearch, #715-585-150) were incubated in blocking buffer for 1 hr at room temperature. Images were acquired with an inverted Leica SP8 single-point scanning confocal microscope and processed using Leica Application Suite X software. Images were scanned at 400 Hz, with a z-step size of 0.5 μm, and a final resolution of 0.15 μm/pixel. To assess fluorescence intensity, mean gray values were measured and background fluorescence was subtracted to obtain the actual fluorescence intensity for each axon.

## Analysis of *Spastin* and *Dpysl5* mRNA in DRG neurons

To assess the levels of *Spastin* and *Dpysl5* mRNA in DRG central and peripheral axons, we performed RNAscope analysis followed by immunohistochemistry as previously described (*Li et al., 2021*). Of note, considering the size of our target sequences (>300 nucleotides), we were only able to design probes for *Spastin* and *Dpysl5* using this assay. Briefly, DRGs were dissected and fixed with 4% paraformaldehyde for 24 hr at 4°C, then moved into cryoprotection solution and embedded in OCT. Longitudinal sections of 12 μm were prepared with a cryostat and stored at –80°C. The RNAscope Intro Pack for Multiplex Fluorescent Reagent Kit v2- Mm (#323136, Advanced Cell Diagnostics) was used. Endogenous peroxidases were quenched, antigens were retrieved, and double-Z probes designed for each transcript were hybridized. The HRP signal was developed for each channel, followed by incubation with Opal Dye fluorophores (Opal Dye 520 to label *Spastin* mRNA [#FP1487001KT, Akoya] and Opal Dye 620 to label *Dpysl5* mRNA [#FP1495001KT, Akoya]). Immunohistochemistry was performed immediately after the RNAscope. Slides were blocked with 10% donkey serum. The primary antibody rabbit anti-myelin basic protein (1:1000, #10458-1-AP, Proteintech, RRID:AB_2250289) was incubated overnight, followed by the secondary antibody Alexa Fluor 647 anti-Rabbit (1:500, #711-605-152, Jackson ImmunoResearch, RRID:AB_2492288) for 1 hr at room temperature. Samples were imaged using an inverted Leica SP8 single-point scanning confocal microscope and acquired using the Leica Application Suit X software. *Spastin* and *Dpysl5* mRNA puncta were quantified using Fiji/ImageJ Cell Counter, and the total axonal mRNA density was determined by dividing the number of mRNA puncta per image area. Three non-consecutive peripheral and central DRG sections per animal were quantified.

## Immunofluorescence of DRG neuron cultures

DRG neurons from E16 Wistar rats at DIV 21 were fixed with pre-warmed 2% PFA diluted in PHEM buffer (65 mM PIPES, 25 mM HEPES, 10 mM EGTA, 3 mM MgCl$_2$, and 0.1% Triton X-100, pH of 6.9) for 20 min. Endogenous autofluorescence and IgG were blocked with ammonium chloride and

M.O.M. blocking reagent (# MKB-2213, Vecto Laboratories, RRID:AB_2336587), respectively. Primary antibodies rabbit anti-polyglutamate chain (1:1000, # AG-25B-0030-C050, Adipogen), mouse anti-acetylated tubulin (1:500, #T7451, Sigma-Aldrich), rabbit anti-Δ2 tubulin (1:1000, #AB3203, Millipore), rabbit anti-βIII-tubulin (1:1000, #302302, Synaptic Systems, RRID:AB_10637424), and mouse anti-βIII-tubulin (1:1000, #G7121, Promega, RRID:AB_430874) were incubated overnight, followed by incubation with the secondary antibodies Alexa Fluor 594 anti-mouse (1:500, #715-585-150, Jackson ImmunoResearch, RRID:AB_2340854), Alexa Fluor 488 anti-rabbit (1:500, #711-545-152, Jackson ImmunoResearch, RRID:AB_2313584), Alexa Fluor 594 anti-rabbit (1:500, #A21207, Invitrogen), Alexa Fluor 647 anti-rabbit (1:500, # 711-605-152, Jackson ImmunoResearch), and DAPI staining. Images were acquired with an inverted Leica SP5 single-point scanning confocal microscope using Leica LasAF software (version 2.6.3.8173), with a pinhole size of 102.9 µm, calculated at 1 AU for 580 nm emission, and a z-step size of 0.3 µm with a final resolution of 0.24 µm/pixel. We assessed the fluorescence intensity of polyglutamylated, acetylated, Δ2 tubulin, and βIII-tubulin at the large and thin axons in a region of 10 µm from the T-junction. This analysis was conducted using ImageJ/Fiji software by calculating their mean gray values, with background fluorescence intensity subtracted. PTMs mean gray values were normalized against βIII-tubulin mean gray values. Eight pseudo-unipolar DRG neurons were quantified per independent experiment. Representative images of various developmental stages of DRG neurons were captured using the inverted Leica SP8 single-point scanning confocal microscope. Images were scanned at 400 Hz, with a final resolution of 0.28 µm/pixel.

## Analysis of regeneration of dorsal column tract axons

Serial sagittal sections of the spinal cord (50 µm) from non-injured mice, spinal cord-injured mice, and mice where the spinal cord injury was preceded by a sciatic nerve injury (conditioning group) were collected for immunofluorescence against cholera toxin-B. The sections were blocked and incubated overnight with goat anti-cholera toxin-B primary antibody (1:30.000, #703, List Biologicals, RRID:AB_10013220), followed by incubation with biotinylated horse anti-goat antibody (1:200, #BA-9500, Vector Laboratories, RRID:AB_2336123) and Alexa Fluor 568-streptavidin (1:1000, #S11226, Invitrogen, RRID:AB_2315774). Cell nuclei were stained with DAPI, and the tissue was imaged using an IN Cell Analyzer 2000 automated microscope (GE Healthcare) in DAPI and FITC channels to visualize nuclei and cholera toxin-B-positive axons, respectively. Mosaics of a single spinal cord were stitched using the IN Cell Developer Toolbox (GE Healthcare) software. Lesion borders were delineated based on nuclei alignment – organized outside the lesion but disordered within the injured area – and further corroborated by cholera toxin-B staining, as most injured central DRG axons halt their growth at the lesion site. To measure the number of regenerating dorsal column tract axons, the lesion border was outlined and the number of cholera toxin-B-positive axons growing inside the lesion area were counted using the Fiji/ImageJ Cell Counter plugin. Were analyzed up to six spinal cord sections per animal. The fluorescence intensity of cholera toxin-B-positive axons was quantified using ImageJ/Fiji software by calculating the mean gray value in different spinal cord sections. Were analyzed up to six spinal cord sections per animal.

## Analysis of myelinated and unmyelinated axons in spastin knockout animals

The sciatic nerve of *Spastin* knockout mice and wild-type littermates was postfixed with 2% osmium tetroxide and embedded in Epon. For the analysis of myelinated axons, cross-sections at 500 nm were stained with 1% paraphenylenediamine (PPD) and mounted in dibutylphthalate polystyrene xylene (DPX). The entire area of the nerve was imaged using an Olympus optical microscope (Olympus, Japan) equipped with an Olympus DP 25 camera, a ×40/0.17 objective, and Cell B software (Olympus). Mosaics with a resolution of 0.08 µm/pixel were stitched in Photoshop (Adobe, USA, RRID:SCR_014199), using the command Photomerge. The Fiji/ImageJ Cell Counter plugin was used to measure the density of myelinated axons. For the analysis of unmyelinated axons, samples were cut at a thickness of 50 nm. Images were acquired using the transmission electron microscope JEOL JEM-1400 (JEOL, Japan) and digitally recorded using a CCD digital camera Orius 1100W (Gatan, USA), with an 8K times magnification and a final resolution of 0.006 µm/pixel. The Fiji/ImageJ Cell Counter plugin was used to determine the density of unmyelinated axons. Only unmyelinated axons

inside Remak bundles were used for quantification. A total of 12 non-overlapping images per animal were quantified.

## Culture of adult DRG neurons on aggrecan

Culture of adult DRG neurons was performed following the protocol described in *Fleming et al., 2009*. Briefly, DRGs from the lumbar root L3, L4, and L5 of adult *Spastin* knockout mice and wild-type littermates were isolated, freed of roots, and digested with 0.125% collagenase-IV-S (Sigma, C1889) for 90 min at 37°C with 5% $CO_2$. After enzymatic digestion, the DRGs were mechanically dissociated using fire-polished Pasteur pipettes of decreasing diameters. The cells were dissociated in culture media consisting of DMEM:F12 (Sigma, D8437) supplemented with 1× B27, 1% penicillin–strepto-mycin, 2 mM L-glutamine, and 50 ng/ml NGF. To separate neurons from glial cells, the dissociated cells were centrifuged in a 15% BSA (Sigma, A3294) gradient for 10 min at $200 \times g$. The neuron-containing pellet was then seeded at 5000 neurons per well in 24-well plates pre-coated with 20 μg/ml poly-L-lysine and 5 μg/ml laminin:20 μg/ml aggrecan (Sigma, A1960). Cultures were maintained at 37°C in a 5% $CO_2$ incubator and fixed in pre-warmed 2% PFA at DIV1. Immunofluorescence for βIII-tubulin was performed as described above. Imaging was conducted using a Leica DMI6000 widefield inverted motorized microscope (Leica Microsystems) equipped with a Hamamatsu FLASH4.0 camera (Hamamatsu, Japan). Samples were excited at 460/40 nm, with an exposure time of 500 ms. Leica Las X software was used for image acquisition, and all images were saved as LIF files with a final resolution of 0.31 μm/pixel. Tiles were acquired and merged using the smooth blend option. The merged images were converted to Imaris files, and the Filaments tool was used to reconstruct and analyze the neurites by performing automatic tracing. To ensure the accuracy of each traced neuron, we manually refined and edited the automatically traced filaments to reduce errors. Five wild-type and five *Spastin* knockout merged images, containing between 10 and 40 neurons each, were quantified. The number of primary neurites was obtained through Sholl analysis.

## Statistical analysis

Depending on the sample size, the normality of the datasets was tested using the Shapiro–Wilk or D'Agostino–Pearson omnibus normality test. If data followed a normal distribution, a two-tailed Student's *t*-test, one-way or two-way ANOVA were used. Student's *t*-test was applied only for comparisons between two groups; a paired *t*-test was used when two variables were measured in a single subject. For multiple comparisons where a single independent variable is present, we utilized one-way ANOVA. Repeated measures were applied when multiple measurements were taken from the same neuron. Two-way ANOVA was used whenever two independent variables were present. ANOVA were followed by a Fisher's least significant difference (LSD) post hoc test. For the data that did not follow a normal distribution, a Wilcoxon matched pair or a Mann–Whitney *U* nonparametric test was used. The chi-square test was applied to compare categorical data. The statistical details, including the statistical test used in each experiment, the exact value of *n*, what *n* represents, the number of animals, and the number of independent experiments used, can be found in the figure legends. All statistical tests and data visualization were performed using GraphPad Prism 8 (GraphPad, RRID:SCR_002798), with results presented as mean ± SEM. p-values are represented by asterisks, where ****p≤0.0001, ***p≤0.001, ** p≤0.01, *p≤0.05, and nonsignificant (ns) p>0.05.

## Acknowledgements

We thank the i3S Advanced Light Microscopy, Histology, Electron Microscopy and BioSciences Screening (PT-OPENSCREEN NORTE-01–0145-FEDER-085468) platforms, all part of PPBI (POCI-01-0145-FEDER-022122). We also thank the i3S Animal and the Cell Culture and Genotyping facilities. We thank the Advanced Imaging Unit at Instituto Gulbenkian de Ciência (PPBI-POCI-01-0145-FEDER-022122/LISBOA-01–0246-FEDER-000037_SingleCell). We are grateful to Sandra Braz for her assistance with performing spinal cord injuries. Illustrations are by Rita Sousa Costa (Aparas Design Lab). This work was funded by national funds through Foundation for Science and Technology (FCT), under the project DRI/India/0336/2020, SFRH/BD/143926/2019, and COVID/BD/153330/2023, the German Research Foundation (GRK1459; KN556/11-1; ID 450131873; TRR 274/1 2020), the Deutsche Gesellschaft für Muskelkranke (LE 4610/1-1), the German Center for Neurodegenerative Diseases, and SyNergy Cluster (EXC 2145; ID 390857198).

## Additional information

### Funding

| Funder | Grant reference number | Author |
| --- | --- | --- |
| Fundação para a Ciência e a Tecnologia | DRI/India/0336/2020 | Monica M Sousa |
| Fundação para a Ciência e a Tecnologia | SFRH/BD/143926/2019 | Ana Catarina Costa |
| Fundação para a Ciência e a Tecnologia | COVID/BD/153330/2023 | Ana Catarina Costa |
| Deutsche Forschungsgemeinschaft | GRK1459 | Matthias Kneussel |
| Deutsche Forschungsgemeinschaft | KN556/11-1 | Matthias Kneussel |
| Deutsche Forschungsgemeinschaft | ID 450131873 | Monika S Brill |
| Deutsche Gesellschaft für Materialkunde | LE 4610/1-1 | Monika S Brill |
| Deutsche Forschungsgemeinschaft | TRR 274/1 2020 | Thomas Misgeld |
| Deutsches Zentrum für Neurodegenerative Erkrankungen | | Thomas Misgeld |
| SyNergy | EXC 2145 | Thomas Misgeld |
| Synergy | ID 390857198 | Thomas Misgeld |

The funders had no role in study design, data collection and interpretation, or the decision to submit the work for publication.

### Author contributions

Ana Catarina Costa, Conceptualization, Data curation, Formal analysis, Validation, Investigation, Visualization, Methodology, Writing – original draft, Writing – review and editing; Blanca R Murillo, Conceptualization, Data curation, Formal analysis, Investigation, Methodology, Writing – review and editing; Rita Bessa, Ricardo Ribeiro, Formal analysis, Methodology; Tiago Ferreira da Silva, Methodology; Patrícia Porfírio-Rodrigues, Gabriel G Martins, Conceptualization, Methodology; Pedro Brites, Conceptualization, Writing – review and editing; Matthias Kneussel, Resources, Funding acquisition, Writing – review and editing; Thomas Misgeld, Monika S Brill, Conceptualization, Resources, Supervision, Funding acquisition, Writing – review and editing; Monica M Sousa, Conceptualization, Resources, Formal analysis, Supervision, Funding acquisition, Writing – original draft, Writing – review and editing

### Author ORCIDs

Ana Catarina Costa ⓘ http://orcid.org/0000-0001-5359-1981
Gabriel G Martins ⓘ https://orcid.org/0000-0002-6506-9776
Matthias Kneussel ⓘ https://orcid.org/0000-0003-4900-366X
Thomas Misgeld ⓘ https://orcid.org/0000-0001-9875-6794
Monika S Brill ⓘ https://orcid.org/0000-0001-5422-9175
Monica M Sousa ⓘ https://orcid.org/0000-0002-4524-2260

### Ethics

Animal experiments followed EU Directive 2010/63/EU and national Decree-law number 113-2013, approved by the i3S Ethical Committee and Portuguese Veterinarian Board.

Reviewer #1 (Public review): https://doi.org/10.7554/eLife.104069.3.sa1
Reviewer #2 (Public review): https://doi.org/10.7554/eLife.104069.3.sa2

Reviewer #3 (Public review): https://doi.org/10.7554/eLife.104069.3.sa3
Author response https://doi.org/10.7554/eLife.104069.3.sa4

## Additional files

### Supplementary files
MDAR checklist

### Data availability
All data generated or analyzed during this study are included in the manuscript and supporting files.

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

# Appendix 1

**Appendix 1—key resources table**

| Reagent type (species) or resource | Designation | Source or reference | Identifiers | Additional information |
|---|---|---|---|---|
| Strain, strain background (*Rattus norvegicus*) | Wistar rats | Charles River | Strain Code 003 | |
| Strain, strain background (*Mus musculus*) | C57BL/6 | Charles River | Strain Code 027 RRID:MGI:2159769 | |
| Strain, strain background (*M. musculus*) | Thy1-EB3-eGFP | From Thomas Misgeld Group | | |
| Strain, strain background (*M. musculus*) | *Spastin* knockout | From Matthias Kneussel Group | | |
| Transfected construct (*M. musculus*) | Lentivirus CMV-EB3-GFP | Plasmid from Frank Polleux | | Lentiviral construct to transduce DRG cells with EB3-GFP. |
| Transfected construct (*M. musculus*) | AAVPHP.S-CMV-eGFP | VectorBuilder | Cat# VB010000-9394npt | AAV construct to transduce DRG cells with eGFP. |
| Transfected construct (*M. musculus*) | Lentivirus synapsin-Tom20-GFP | Charité NeuroCure, University of Berlin | Cat# BLV-569a | Lentiviral construct to transduce DRG cells with Tom20-GFP. |
| Antibody | Anti-tyrosinated tubulin [YL1/2] (rat monoclonal) | Abcam | Cat# ab6160 RRID:AB_305328 | WB (1:5000) |
| Antibody | Anti-acetylated tubulin (mouse monoclonal) | Sigma-Aldrich | Cat# T7451 RRID:AB_609894 | WB (1:10,000) |
| Antibody | Anti-Δ2 tubulin (rabbit polyclonal) | Millipore | Cat# AB3203 RRID:AB_177351 | IF cells (1:1000) WB (1:2000) |
| Antibody | Anti-polyglutamylated tubulin (rabbit polyclonal) | Adipogen | Cat# AG-25B-0030-C050 | IF cells (1:1000) WB (1:10,000) |
| Antibody | Anti-spastin (mouse monoclonal) | Santa Cruz | Cat# sc-81624 RRID:AB_2286628 | WB (1:500) |
| Antibody | Anti-katanin p60 (rabbit polyclonal) | Proteintech | Cat# 17560-1-AP RRID:AB_10694670 | IF tissue (1:500) WB (1:500) |
| Antibody | Anti-tau (mouse monoclonal) | Cell Signalling | Cat# 4019 | IF tissue (1:100) WB (1:1000) |
| Antibody | Anti-CRMP5 (rat monoclonal) | Santa Cruz | Cat# sc-58515 RRID:AB_782270 | IF tissue (1:200) WB (1:200) |
| Antibody | Anti-vinculin (rabbit monoclonal) | Invitrogen | Cat# 700062 | WB (3:10,000) |
| Antibody | Anti-CT-B primary (goat polyclonal) | List Biologicals | Cat# 703 RRID:AB_10013220 | IF tissue (1:30,000) |
| Antibody | Biotinylated horse anti-(goat polyclonal) | Vector Laboratories | Cat# BA-9500 RRID:AB_2336123 | IF tissue (1:30,000) |

*Appendix 1 Continued*

| Reagent type (species) or resource | Designation | Source or reference | Identifiers | Additional information |
|---|---|---|---|---|
| Antibody | Anti-myelin basic protein (rabbit polyclonal) | Proteintech | Cat# 10458-1-AP RRID:AB_2336123 | IF tissue (1:1000) |
| Antibody | Anti-βIII-tubulin (rabbit polyclonal) | Synaptic Systems | Cat# 302302 RRID:AB_10637424 | IF tissue (1:200) IF cells (1:1000) |
| Antibody | Anti-βIII-tubulin (mouse monoclonal) | Promega | Cat# G7121 RRID:AB_430874 | IF tissue (1:500) IF cells (1:1000) |
| Antibody | Alexa Fluor 647 anti- (rabbit polyclonal) | Jackson ImmunoResearch | Cat# 711-605-152 RRID:AB_2492288 | IF tissue and cells (1:500) |
| Antibody | Alexa Fluor 594 anti- (mouse polyclonal) | Jackson ImmunoResearch | Cat# 715-585-150 RRID:AB_2340854 | IF tissue (1:250) IF cells (1:500) |
| Antibody | Alexa Fluor 488 anti- (rabbit polyclonal) | Jackson ImmunoResearch | Cat# 711-545-152 RRID:AB_2313584 | IF tissue (1:250) IF cells (1:500) |
| Antibody | Alexa Fluor 594 anti- (rabbit polyclonal) | Invitrogen | Cat# A21207 | IF cells (1:500) |
| Sequence-based reagent | RNAscope Probe Mm-Dpysl5-C2 | Advanced Cell Diagnostics | Cat# 1112131-C2 | |
| Sequence-based reagent | RNAscope Probe Mm-Spast | Advanced Cell Diagnostics | Cat# 849171 | |
| Chemical compound, drug | Cholera toxin-B | List Biologicals | Cat# 103B | |
| Chemical compound, drug | Protease inhibitors | Roche | Cat# 04693124001 | |
| Chemical compound, drug | 4–20% acrylamide gels | Bio-Rad | Cat# 5671093 | |
| Chemical compound, drug | Nitrocellulose membranes | GE Healthcare Life Sciences | Cat# 10600013 | |
| Chemical compound, drug | M.O.M. blocking reagent | Vector Laboratories | Cat# MKB-2213, RRID:AB_2336587 | |
| Commercial assays or kit | Detergent Compatible Protein Assay | Bio-Rad | Cat# 5000116 | |
| Commercial assays or kit | RNAscope Intro Pack for Multiplex Fluorescent Reagent Kit v2- Mm | Advanced Cell Diagnostics | Cat# 323136 | |
| Software, algorithm | Image Lab Software for PC Version 6.1 | Bio-Rad | RRID:SCR_014210 | |
| Software, algorithm | IQ 2 software | Andor Technology | N/A | |
| Software, algorithm | Fiji/ImageJ 1.53t software | NIH | RRID:SCR_002285 | |
| Software, algorithm | Leica Application Suit X software | Leica | RRID:SCR_013673 | |
| Software, algorithm | Leica LasAF software Version 2.6.3.8173 | Leica | N/A | |
| Software, algorithm | Slidebook software Version 2022 | 3i | RRID:SCR_014423 | |

*Appendix 1 Continued*

| Reagent type (species) or resource | Designation | Source or reference | Identifiers | Additional information |
|---|---|---|---|---|
| Software, algorithm | IN Cell Developer Toolbox software | GE Healthcare | N/A | |
| Software, algorithm | Cell B software | Olympus | N/A | |
| Software, algorithm | Adobe Photoshop | Adobe | RRID:SCR_014199 | |
| Software, algorithm | GraphPad Prism 8 | GraphPad | RRID:SCR_002798 | |
| Other | PureBlu DAPI Nuclear Staining Dye | Bio-Rad | Cat# 1351303 | Highly pure formulation of DAPI (4',6-diamidino-2-phenylindole), a well-characterized blue-emitting fluorescent compound widely utilized for nuclear staining |
| Other | Alexa Fluor 568-streptavidin | Invitrogen | Cat# S11226 RRID:AB_2315774 | Streptavidin covalently attached to a fluorescent label (Alexa Fluor dye) IF tissue (1:1000) |
| Other | Opal Dye 520 | Akoya | Cat# FP1487001KT | Opal dyes are fluorescent dyes commonly used in multiplex immunofluorescence in a single tissue sample by using tyramide signal amplification IF tissue (1:750) |
| Other | Opal Dye 620 | Akoya | Cat# FP1495001KT | Opal dyes are fluorescent dyes commonly used in multiplex immunofluorescence in a single tissue sample by using tyramide signal amplification IF tissue (1:750) |
| Other | Glass-bottom 4-well chambered coverslips | Ibidi | Cat# 80427 | Chambered coverslip with 1.5H glass bottom enabling live cell imaging |
| Other | Glass-bottom 35 mm imaging dish | Ibidi | Cat# 81158 | 1.5H glass bottom used for live imaging |
| Other | Micro-feather ophthalmic scalpel | Delta Microscopes | Cat# 72045-45 | High-precision surgical blade with 45° used to perform spinal cord dorsal hemisection |
| Other | Fuji Medical X-Ray Film | Fujifilm | Cat# 16195209 | Blue sensitive universal film used for imaging chemiluminescent signals in western blot membranes |
| Other | Molecular Imager GS800 | Bio-Rad | Cat #170-7980 | Densitometer used for imaging and quantifying protein bands on X-ray films |
| Other | ChemiDoc Imaging System | Bio-Rad | Cat# 12003153 | Allows sensitive imaging of chemiluminescent western blots membranes |

