## [Editor Report · eLife Assessment]

In their **important** article, Costa et al. establish an in vitro model for dorsal root ganglion (DRG) axonal asymmetry, revealing that central and peripheral axon branches have distinct patterns of microtubule populations that are linked to their differential regenerative capacities. The authors employ creative tissue culture methods to demonstrate how these branches develop uniquely in vitro, offering a potential explanation for long-observed regeneration disparities. The **convincing** evidence provides a contribution to our understanding of the neuronal cytoskeleton and axonal regeneration.

---

## [Referee Report · Reviewer #1 (Public review)]

Summary:

This paper describes a new in vitro model for DRG neurons that recapitulates several key differences between the peripheral and central branches of DRG axons in vivo. These differences include morphology (with one branch being thinner than the other), and regenerative capacity (with the peripheral branch displaying higher regenerative capacity). The authors analyze the abundance of various microtubule associated protein (MAPs) in each branch, as well as the microtubule dynamics in each branch and find significant differences between branches. Importantly, they found that a well-known conditioning paradigm (prior lesion of the peripheral branch improves the regenerative capacity of the central branch) is not only reproduced in this system, but also leads to loss of the asymmetry of MAPs between branches. Zooming in on one MAP that shows differential abundance between the axons, they find that the severing enzyme Spastin is required for the asymmetry in microtubule dynamics and in regenerative capacity following a conditioning lesion

Strengths:

The establishment of an experimental system that recapitulates DRG axon asymmetry in vitro is an important step that is likely to be useful for other studies. In addition, identifying key molecular signatures that differ between central and peripheral branches, and determining how they are lost following a conditioning lesion adds to our understanding of why peripheral axons have a better regenerative capacity. Last, the authors use of an in vivo model system to support some of their in vitro findings is a strength of this work.

Weaknesses:

One weakness of the manuscript is that to a large degree, one of its main conclusions (MAP symmetry underlies differences in regenerative capacity) relies mainly on a correlation, without firmly establishing a causal link. However, this is weakness is relatively minor because (1) it is partially addressed with the Spastin KO and (2) there isn't a trivial way to show a causal relationship in this case. (3) It is addressed in the Discussion section.

---

## [Referee Report · Reviewer #2 (Public review)]

Summary:

The authors set out to develop a tissue culture method in which to study the different regenerative abilities of the central and peripheral branch of sensory axons. Neurons developed a small and large branch, which have different regenerative abilities, different transport rates and different microtubule properties. The study provides convincing evidence that the two axonal branches differ in a way to corresponds to in vivo. The different regenerative abilities of the two branches are an important observation, because until now it has not been clear whether this difference is intrinsic to the neuron and axons or due to differences in the environment surrounding the axons. The authors have then looked for molecular explanations of the differences between the branches. They find different transport rates and different microtubule dynamics. The different microtubule dynamics are explained by differing levels of spastin, an enzyme that severs microtubules encouraging dynamics.

Strengths:

The differences between the two branches are clearly shown, together with differences in transport, microtubule dynamics and regeneration. The in vitro model is novel and could be widely used. The methods used are robust and generally accepted.

Weaknesses:

The revised version of the paper has addressed the weaknesses that were identified.

---

## [Referee Report · Reviewer #3 (Public review)]

Summary:

In this manuscript, Costa and colleagues investigate how asymmetry in dorsal root ganglion (DRG) neurons is established. The authors developed an in vitro system that mimics the pseudo-unipolar morphology and asymmetry of DRG neurons during the regeneration of the peripheral and central branch axons. They suggest that central-like DRG axons exhibit a higher density of growing microtubules. By reducing the polymerization of microtubules in these central-like axons, they were able to eliminate the asymmetry in DRG neurons.

Strengths:

The authors point out a distinct microtubule-associated protein signature that differentiates between DRG neurons' central and peripheral axonal branches. Experimental results demonstrate that genetic deletion of spastin eliminated the differences in microtubule dynamics and axon regeneration between the central and peripheral branches.

Weaknesses:

While some of the data are compelling, experimental evidence does not fully support the main claims.

In its current form, the study is primarily descriptive and lacks convincing mechanistic insights. It misses important controls and further validation using 3D in vitro models.

The significance of studying microtubule polymerization to DRG asymmetry in vitro is questionable, especially considering the model's validity. Classifying the central and peripheral-like branches in cultured DRG neurons will require further in-depth characterization. Additional validation using adult DRG neuron cultures not aged in vitro will be required in future studies.

The comparison of asymmetry associated with a regenerative response between in vitro and in vivo paradigms has significant limitations due to the nature of the in vitro culture system. When cultured in isolation, DRG neurons fail to form functional connections with appropriate postsynaptic target neurons (the central branch) or to differentiate the peripheral domains associated with the innervation of target organs. Rather than growing neurons on a flat, hard surface like glass, more physiologically relevant substrates and/or culturing conditions should be considered. This approach could help eliminate potential artifacts caused by plating adult DRG neurons on a flat surface. Additionally, the authors should consider replicating their findings in a 3D culture model or using dorsal root ganglia explants, where both centrally and peripherally projecting axons are present.

Panels 5H-J require additional processing with astrocyte markers to accurately define the lesion borders. Furthermore, including a lower magnification would facilitate a direct comparison of the lesion site. The use of cholera toxin subunit B (CTB) to trace dorsal column sensory axons is prone to misinterpretation, as the tracer accumulates at the axon's tip. This limitation makes it extremely challenging to distinguish between regenerating and degenerating axons.

---

## [Author Response]

The following is the authors’ response to the original reviews.

**Public Reviews**

**Reviewer #1 (Public review)**
Weaknesses:The main weakness of the manuscript is that to a large degree, one of its main conclusions (MAP symmetry underlies differences in regenerative capacity) relies mainly on a correlation, without firmly establishing a causal link. However, this weakness is relatively minor because (1) it is partially addressed with the Spastin KO and (2) there isn't a trivial way to show a causal relationship in this case.

We thank Reviewer #1 for their positive assessment of our manuscript. To further strengthen the claim that MAP asymmetry underlies differences in regenerative capacity, we could investigate the effect of depleting other MAPs that lose asymmetry after conditioning lesion (CRMP5 and katanin). One would expect that similarly to spastin, this would disrupt the physiological asymmetry of DRG axons and impair axon regeneration. We further discussed this issue in the revised version of the manuscript (page 17, line 381).

**Reviewer #2 (Public review)**
Weaknesses:In order for the method to be used it needs to be better described. For instance what proportion of neurons develop just two axonal branches, one of which is different? How selective are the researchers in finding appropriate neurons?

We thank Reviewer #2 for their positive assessment of our manuscript. As suggested, we included further methodological details on the in vitro system in the revised version of the manuscript. We have previously evaluated the percentage of DRG neurons exhibiting different morphologies in our cultures: multipolar (4±1%), bipolar, (35±8%) bell-shaped (17±5%), and pseudo-unipolar neurons (43±3%). This was included in the revised manuscript on Figure 1B and page 5, line 107. All the pseudo-unipolar neurons analysed had distinct axonal branches in terms of diameter and microtubule dynamics. For imaging purposes, we selected pseudounipolar neurons with axons unobstructed from other cells or neurites within a distance of at least 20–30 μm from the bifurcation point, to ensure optimal imaging. In the case of laser axotomy experiments, this distance was increased to 100–200 μm to ensure clear analysis of regeneration. These selection criteria is now detailed in the Methods (page 19, line 417, and page 21, line 474).

**Reviewer #3 (Public review):**
(1) Weaknesses:While some of the data are compelling, experimental evidence only partially supports the main claims. In its current form, the study is primarily descriptive and lacks convincing mechanistic insights. It misses important controls and further validation using 3D in vitro models.

We recognize the importance of further exploring the contribution of other MAPs to microtubule asymmetry and regenerative capacity of DRG axons. In future work, we plan to investigate this issue using knockout mice for katanin and CRMP5. Regarding the mechanisms underlying the differential localization of proteins in DRG axons, we performed *in-situ* hybridization to evaluate the availability of axonal mRNA but no differences were found between central and peripheral DRG axons (Figure 4 – figure supplement 2). To address whether differences in protein transport exist, we attempted to transduce DRG neurons with GFP-tagged spastin both in vitro and in vivo. However, these experiments were inconclusive as very low levels of spastin-GFP were detected. We are actively optimizing these approaches and will address this challenge in future studies. These points were further discussed in the revised manuscript (page 15, line 330 and page 17, line 381).

(2) Given the heterogeneity of dorsal root ganglion (DRG) neurons, it is unclear whether the in vitro model described in this study can be applied to all major classes of DRG neurons.

We acknowledge the diversity of DRG neurons and agree that assessing the presence

of different DRG subtypes in our culture system will enrich its future use. Despite this heterogeneity, we focused on DRG neuron features that are common to all subtypes i.e, pseudo-unipolarization and higher regenerative capacity of peripheral branches. This point was addressed on page 14, line 309 of the revised manuscript.

(3) Also unclear is the inconsistency with embryonic DRG cultures with embryonic (E)16 from rats and E13 from mice (spastin knockout and wild-type controls).

Given our previous experience in establishing DRG neuron cultures from E16 Wistar rats and E13 C57BL/6 mice, these developmental stages are equivalent, yielding cultures of DRG neurons with similar percentages of different morphologies. Of note, in our colonies, gestation length is ~19 days in C57BL/6 mice (background of the *spastin* knockout line) and ~22 days in Wistar Han rats. This was further clarified in the Methods (page 18, line 404).

(4) Furthermore, the authors stated (line 393) that only a small subset of cultured DRG neurons exhibited a pseudo-unipolar morphology. The authors should include the percentage of the neurons that exhibit a pseudo-unipolar morphology.

We have previously evaluated the percentage of DRG neurons exhibiting different morphologies in our cultures: multipolar (4±1%), bipolar, (35±8%) bell-shaped (17±5%), and pseudo-unipolar neurons (43±3%). This was included in the revised manuscript on Figure 1B and on page 5, line 107. In line 393, we referred specifically to an experimental setup where DRG neuron transduction was done, and 30 transduced neurons were randomly selected for longitudinal imaging. From these, the number of viable pseudo-unipolar DRG neurons was limited by both the random nature of viral transduction and light-induced toxicity throughout continuous imaging over seven consecutive days at hourly intervals. This was clarified in the revised manuscript (page 20, line 438).

(5) The significance of studying microtubule polymerization to DRG asymmetry in vitro is questionable, especially considering the model's validity. The authors might consider eliminating the in vitro data and instead focus on characterizing DRG asymmetry in vivo both before and after a conditioning lesion. If the authors choose to retain the in vitro data, classifying the central and peripheral-like branches in cultured DRG neurons will require further in-depth characterization. Additional validation should be performed in adult DRG neuron cultures not aged in vitro.

The in vitro system here presented reliably reproduces several key features of DRG neurons observed in vivo*,* including asymmetry in axon diameter, regenerative capacity, axonal transport, and microtubule dynamics. Of note, most studies in the field have been done using multipolar DRG neurons that do not recapitulate in vivo morphology and asymmetries. Thus, the current in vitro model serves as a versatile tool for advancing our understanding of DRG biology and associated diseases. This system is particularly suited to study axon regeneration asymmetries, and enables the investigation of mechanisms occurring at the stem axon bifurcation, such as asymmetric protein transport and microtubule dynamics, which are challenging to examine in vivo due to the length of the stem axon and the difficulty of locating the DRG T-junction. It will be important to optimize similar cultures using adult DRG neurons. However, this comes with challenges, such as lower cell viability. This is the case with multiple other neuron types for which the vast majority of cultures are obtained from embryonic tissue. These concerns were addressed in the revised version of the manuscript (page 13, line 296 and page 14 line 302).

(6) The comparison of asymmetry associated with a regenerative response between in vitro and in vivo paradigms has significant limitations due to the nature of the in vitro culture system. When cultured in isolation, DRG neurons fail to form functional connections with appropriate postsynaptic target neurons (the central branch) or to differentiate the peripheral domains associated with the innervation of target organs. Rather than growing neurons on a flat, hard surface like glass, more physiologically relevant substrates and/or culturing conditions should be considered. This approach could help eliminate potential artifacts caused by plating adult DRG neurons on a flat surface. Additionally, the authors should consider replicating their findings in a 3D culture model or using dorsal root ganglia explants, where both centrally and peripherally projecting axons are present.

We agree that a more sophisticated system, such as a compartmentalized culture, holds great potential for future research. In this respect, we are currently engaged in developing such models. A compartmentalized system would enable the separation of three compartments: central nervous system neurons, DRG neurons, and peripheral targets. While previous efforts to create compartmentalized DRG cultures have been reported (e.g., PMID: 11275274 and PMID: 37578145), these systems have not demonstrated the development of pseudo-unipolar morphology. Incorporating non-neuronal DRG cells into the DRG neuron compartment, may successfully support the development of a pseudo-unipolar morphology.

We also recognize the importance of dimensionality in fostering pseudo-unipolar morphology. Of note, our model provides a 3D-like environment, as DRG glial cells are continuously replicating over the 21 days in culture. In relation to DRG explants, we attempted their use but encountered limitations with confocal microscopy as the axial resolution was insufficient to resolve processes at the DRG T-junction or within individual branches. The above issues are now discussed in the revised manuscript (page 14, line 312).

(7) Panels 5H-J require additional processing with astrocyte markers to accurately define the lesion borders. Furthermore, including a lower magnification would facilitate a direct comparison of the lesion site.

In our study, we relied on the alignment of nuclei to delineate the lesion site as in our accumulated experience, this provides an accurate definition of the lesion boarder. Outside the lesion, the nuclei are well-aligned, while at the lesion site, they become randomly distributed. Additionally, CTB staining further supports the identification of the rostral boarder of the lesion, as most injured central DRG axons stop their growth at the injury site. This was further detailed in the Methods of the revised manuscript (page 32, line 730).

(8) The use of cholera toxin subunit B (CTB) to trace dorsal column sensory axons is prone to misinterpretation, as the tracer accumulates at the axon's tip. This limitation makes it extremely challenging to distinguish between regenerating and degenerating axons.

While alternative methods to trace or label regenerating axons exist, CTB is a wellestablished and widely used tracer for central sensory projections, as shown in different studies (PMID: 22681683, PMID: 26831088 and PMID: 33349630). Regarding the concern of possiblebCTB labeling in degenerating axons, we believe this is unlikely to be the case in our system, as in spinal cord injury controls, CTB-positive axons are nearly absent. Also, as regeneration was investigated six weeks after injury, axon degeneration has most likely already occurred as shown in (PMID: 15821747 and PMID: 25937174).

**Recommendations for the authors:**

**Reviewer #1:**
(1) Figure 1 can be improved by adding a quantification of the fraction of neurons at each stage as a function of time.

We have updated Figure 1 to include the quantification of the percentages of different DRG neuron morphologies at DIV21 (Figure 1B), which corresponds to the stage at which all in vitro experiments were conducted.

(2) Figure 3A: why are retrograde transport events not shown?

Retrograde transport events are not displayed as results did not reach statistical significance.

(3) Figure 3 and 4: Combine the quantifications of with/without lesion, such that not only the differences between branches are apparent, but also the differences induced in each branch by the lesion.

As requested, only combined quantifications of microtubule dynamics for naive and conditioning lesion are provided in the revised version of Figure 3 (Figures 3H and 3K), to highlight both branch-specific differences and lesion-induced changes. However, for Figure 4, as the western blots for naive and conditioning lesion were performed on separate gels, it is unfeasible to combine their quantification.

(4) Figure 5: does spastin KO lead to a difference in the "MAP signature" of each branch? Also, if in addition to MAPs there are other known molecules (and an antibody is available) that show differential localization to peripheral/central branches, it would be nice to check if this asymmetry is also lost in spastin KO.

Evaluating the MAP signature in DRG axons from spastin KO mice will be important to explore in future experiments. Despite some scattered reports in the literature, our study is the first to identify a distinct protein signature of central and peripheral DRG axons. This is especially relevant in the case of Tau, as irrespective of the experimental conditions, its levels are always increased in the peripheral DRG axon.

**Reviewer #2:**
(1) Please provide a more complete description of the culture method. Do all neurons develop two asymmetric branches or just a few, and how are they selected? Does the timing of the events in vitro correspond with what is happening to the neurons in embryos?

We have included the percentages of the various DRG neuron morphologies at DIV21 in the revised manuscript (Figure 1B and on page 5, line 107). Additionally, a more detailed description of the culture method is now provided in the Methods, including the criteria used to select pseudo-unipolar neurons (page 19, line 417, and page 21, line 474).

Regarding the timing of events, upon DRG dissociation, neurons reinitiate polarization, taking 21 days to reach approximately 40% pseudo-unipolar morphology. A similar percentage is reached at E16.5 during rat development in vivo (PMID: 8729965).

(2) Are the neurons and their branches resting on the glia? Is there any relation to the presence of glia and the type of growth that is seen?

Yes, neurons and their branches rest on glia. This is required for DRG pseudounipolarization. In future studies, we plan to further investigate neuron-extrinsic mechanisms leading pseudo-unipolarization, and to identify the specific glial cell type(s) needed throughout this process. This is now discussed in the revised manuscript (page 14, line 306).

(3) Is it possible to trace microtubules so as to see whether the microtubules of the two branches mix, or whether they remain separate all the way to the cell bodies?

We used DRG neurons transduced with EB3-GFP, to examine microtubule polymerization at the T-junction through live imaging. This revealed a high continuum of polymerization from the stem axon to the central-like axon (Figure 4 – figure supplement 2D-G). To further determine whether microtubules from both branches mix or remain separate, alternative techniques such as FIB-SEM could be performed. This point is now further discussed in the revised manuscript (page 16, line 352).

(4) Using the term MAPs would lead readers to expect to see an analysis of different levels of MAP1, MAP2, etc. It would be interesting to see this if the authors have done it, but it is not necessary for the paper.

We assessed the expression of MAP2 via western blot in DRG peripheral and central axons and no differences were found. This is now referred to in the Discussion (pages 15, line 327).

(5) The regeneration experiments on the spastin knockouts are complicated by the lesion being in CNS tissue, which introduces various issues. Is there a difference in regeneration after dorsal root crush?

We have not yet examined whether regeneration differs after dorsal root crush in the spastin knockout model. However, this presents an interesting question, as Schwann cells in the dorsal root, may support regeneration of central DRG axons.

**Reviewer #3:**
The authors stated that the normality of the datasets was tested using the Shapiro-Wilk or D'Agostino-Pearson omnibus normality test. Given the low sample size (n = 4) for some of the experiments presented (e.g., Figure 3B), it is not clear how normality was assessed which justifies the use of parametric tests.

We followed GraphPad’s recommendations for selecting the appropriate normality test (https://www.graphpad.com/support/faqid/959/). The D'Agostino-Pearson omnibus K2 test, recommended for its versatility, was used when sample size was 8 or more. For smaller sample sizes (n < 8), we used the Shapiro-Wilk test, which is also widely used in biological research and can be employed with datasets of at least 3 values. These tests guided our decision-making regarding the use of parametric or non-parametric statistical tests.